# Essentiality of fatty acid synthase in the 2D to anchorage-independent growth transition in transforming cells

Maria J. Bueno[1], Veronica Jimenez-Renard[1], Sara Samino[2,3], Jordi Capellades[2,3], Alejandra Junza[2,3], María Luz López-Rodríguez[4], Javier Garcia-Carceles [4], Irene Lopez-Fabuel [5,6], Juan P. Bolaños [5,6], Navdeep S. Chandel [7], Oscar Yanes [2,3], Ramon Colomer [8] & Miguel Quintela-Fandino[1,9,10]*

Upregulation of fatty acid synthase (FASN) is a common event in cancer, although its mechanistic and potential therapeutic roles are not completely understood. In this study, we establish a key role of FASN during transformation. FASN is required for eliciting the anaplerotic shift of the Krebs cycle observed in cancer cells. However, its main role is to consume acetyl-CoA, which unlocks isocitrate dehydrogenase (IDH)-dependent reductive carboxylation, producing the reductive power necessary to quench reactive oxygen species (ROS) originated during the switch from two-dimensional (2D) to three-dimensional (3D) growth (a necessary hallmark of cancer). Upregulation of FASN elicits the 2D-to-3D switch; however, FASN's synthetic product palmitate is dispensable for this process since cells satisfy their fatty acid requirements from the media. In vivo, genetic deletion or pharmacologic inhibition of FASN before oncogenic activation prevents tumor development and invasive growth. These results render FASN as a potential target for cancer prevention studies.

[1] Breast Cancer Clinical Research Unit, CNIO – Spanish National Cancer Research Center, Madrid, Spain. [2] Metabolomics Platform, Department of Electronic Engineering, Universitat Rovira i Virgili, Tarragona, Spain. [3] Biomedical Research Center in Diabetes and Associated Metabolic Disorders, CIBERDEM, Madrid, Spain. [4] Quimica Organica I, Universidad Complutense, Madrid, Spain. [5] Institute of Functional Biology and Genomics (IBFG), Universidad de Salamanca, CSIC, Salamanca, Spain. [6] Centro de Investigación Biomédica en Red sobre Fragilidad y Envejecimiento Saludable (CIBERFES), Institute of Biomedical Research of Salamanca, 37007 Salamanca, Spain. [7] Department of Medicine, Northwestern University Feinberg School of Medicine Chicago, Chicago, IL, USA. [8] Medical Oncology Hospital, Universitario La Princesa, Madrid, Spain. [9] Medical Oncology Hospital, Universitario Quiron, Pozuelo de Alarcon – Madrid, Spain. [10] Medical Oncology, Hospital Universitario de Fuenlabrada, Fuenlabrada – Madrid, Spain. *email: mquintela@cnio.es

Fatty acid synthase (FASN) is a multienzyme protein that catalyzes the synthesis of fatty acids (mainly the long-chain saturated fatty acid palmitate) from acetyl-CoA and malonyl-CoA in the presence of nicotinamide adenine dinucleotide phosphate reduced form (NADPH)[1]. Most adult tissues satisfy their fatty acid needs by uptaking circulating fatty acids from the bloodstream[1,2]. A constant fatty acid supply to the bloodstream is generated by nutrition and lipid storage in adipocytes and hepatocytes[2]. A common feature of tumors from different lineages is FASN overexpression[3–9]. A transgenic mouse model that overexpresses FASN in prostate tissue showed tumor-promoting properties of this enzyme[10]. Constitutive FASN knockout is lethal in the early embryonic stages[11]. Regarding the adult animal, several studies by Semenkovich and colleagues[12–17] demonstrated the involvement of FASN in several physiologic processes. However, conditional deletion is only lethal in the case of the gastrointestinal tract[18]. Taken together, these findings indicate a potential oncogenic role of FASN and a potential high therapeutic index due to the lack of essentiality in most normal tissues.

The effects of FASN inhibition in developed tumors have been well studied. Early inhibitors such as cerulenin, C75, or orlistat only showed modest antitumor effects[8,19–25]. Similar efficacy was observed with second-generation, non-clinical-grade FASN inhibitors[26,27] or gene-silencing approaches[24,28,29]. Although mature data from the first clinical-grade inhibitor are not available, preliminary results do not indicate that targeting FASN may eradicate established tumors[30].

Conversely, the role of FASN could be different during the carcinogenesis process. Complex metabolic rearrangements occur during the transition from normal to malignant cells[31]. Such events require tight regulation, and changes in FASN activity might cause interferences due to the following reasons: (1) FASN activity accounts for the highest cell consumption of NADPH[1], a key regulator of the REDOX balance, which is altered but highly regulated in cancer cells[32]; (2) the ability to grow in three dimensions (3D) in the absence of intercellular matrix attachment is a necessary hallmark that transforming cells must acquire to develop clinical tumors[33], and requires quenching an excess of reactive-oxygen species (ROS)[34]; 3) FASN also consumes acetyl-CoA, which in turn regulates the entrance of glycolysis-derived pyruvate in the Krebs cycle[35]. The anaplerotic shift of the Krebs cycle is also one key characteristic of cancer-rewired metabolism[31]. Since tumor cells can satisfy their fatty acid needs by active uptake from the bloodstream[36], we hypothesized that FASN activity could be essential during the initial steps of the transformation process and thus be a target for cancer prevention. Although in established tumors FASN upregulation has been described as a "cancer autonomous" system to generate its own lipid pools[1,37,38], in the proposed model the requirement of FASN by transforming cells would be unrelated to the enzymatic product.

To study the role of FASN and its involvement in the metabolic reprogramming events that occur during transformation, we designed several in vitro and in vivo models to temporally and spatially control FASN expression. Across models driven by diverse oncogenic hits such as HER2, KRAS, or PyMT (polyomarivurs middle T antigen), we showed how FASN is essential for sustaining the isocitrate dehydrogenase 1 (IDH1)-dependent reductive carboxylation of glutamine[39,40] that allows quenching excessive ROS produced during the transition from two-dimensional (2D) growth to 3D anchorage-independent growth, a hallmark of cancer[33]. The presence of FASN synthetic product in the system does not modify this essentiality. The in vivo implications of these effects in different mouse models were prevention of tumorigenesis.

## Results

**FASN is essential during the transformation process.** We developed a FASN conditional allele where exons 4–8 were flanked by loxP sites (FASN$^{lox}$). We used this allele in combination with a 4-hydroxytamoxifen-inducible Cre recombinase under the control of the ubiquitin C promoter (UBC-CreER$^{T2}$)[41] (Supplementary Fig. 1a). We isolated and immortalized mouse embryonic fibroblasts (MEFs) from FASN$^{lox/lox}$ mice. FASN$^{lox/lox}$ MEFs were infected with retroviral particles expressing the PyMT breast cancer oncogene, KRAS (G12D), or HER2 (A775_G776insYVMA)[42] (FASN$^{lox/lox}$-PyMT, FASN$^{lox/lox}$-KRAS, and FASN$^{lox/lox}$-HER2, respectively; Supplementary Fig. 1b). Colony formation assay showed that the three oncogenes transformed FASN$^{lox/lox}$ MEFs (Fig. 1a and Supplementary Fig. 1c), but only in the presence of palmitate (Supplementary Fig. 4f). When FASN$^{lox/lox}$ MEFs were infected with adenoviral particles expressing Cre to eliminate the conditional FASN allele prior to PyMT, KRAS (G12D), or HER2 (A775_G776insYVMA) expression (FASN$^{\Delta/\Delta}$-PyMT, FASN$^{\Delta/\Delta}$-KRAS, and FASN$^{\Delta/\Delta}$-HER2, respectively), virtually no colonies were recovered (Fig. 1a and Supplementary Fig. 1c). The same inability to form colonies was observed when wild-type, immortalized murine breast epithelial cells HC11 were compared with clustered regularly interspaced short palindromic repeat (CRISPR)-deleted FASN variants transfected with PyMT, KRAS (G12D), or HER2 (Supplementary Fig. 1e–g). Interestingly, the few colonies recovered from the FASN$^{\Delta/\Delta}$-PyMT soft agar culture preserved important residual FASN messenger RNA (mRNA) and protein levels (Fig. 1b), indicating the colony formation process occurred only in "escapers" (i.e., due to the <100% efficient adenovirus-expressing Cre-recombinase (adeno-Cre) infection process). Despite the different ability to form colonies in soft agar, proliferation and apoptotic rates were unaffected by FASN depletion in 2D cultures (Fig. 1c, d, d). Although FASN$^{\Delta/\Delta}$ clones were unable to form colonies, it is important to mention that minor FASN-level fluctuations observed among different FASN$^{lox/lox}$-PyMT clones were not related to an impaired or improved clonogenic ability (Supplementary Fig. 2a, d); similar observations applied for FASN$^{lox/lox}$-KRAS and FASN$^{lox/lox}$-HER2 (Supplementary Fig. 2b, c, e, f). Finally, although non-statistically significant, the lower replication rate of FASN$^{\Delta/\Delta}$ clones in 2D cultures could account for a decreased integration of retroviral particles containing the studied oncogenes, but the examination of PyMT, KRAS, and HER2 levels across different FASN$^{\Delta/\Delta}$ and FASN$^{lox/lox}$ clones did not reveal significant differences (Supplementary Fig. 2g).

**FASN loss impairs metabolic changes during transformation.** Phosphoinositide-3-kinase-protein kinase B/Akt (PI3K-AKT) and mitogen-activated protein kinase (MAPK) hyperactivation are the most frequent oncogenic alterations found in human tumors regardless of the lineage of origin[43]. PyMT-transformed cells show hyperactive PI3K-AKT and MAPK pathways and a strong Warburg effect[44–48]. KRAS mutations and HER2 signaling are also associated with similar metabolic traits[47,49]. We hypothesized that the lack of FASN would impair glycolysis and the anaplerotic shift of the Krebs cycle[31] based on the scheme shown in Fig. 2a.

Congruently with the lack of FASN, we observed an increase in acetyl-CoA and NADPH in FASN$^{\Delta/\Delta}$-PyMT compared with FASN$^{lox/lox}$-PyMT MEFs (Fig. 2b), which was coupled to a >6-fold decrease in pyruvate dehydrogenase activity (Supplementary Fig. 3a). By using an in vitro assay with mitochondrial lysates (i.e., in conditions where the original mitochondrial stoichiometry is lost), we observed a trend to phenocopy such inhibition by

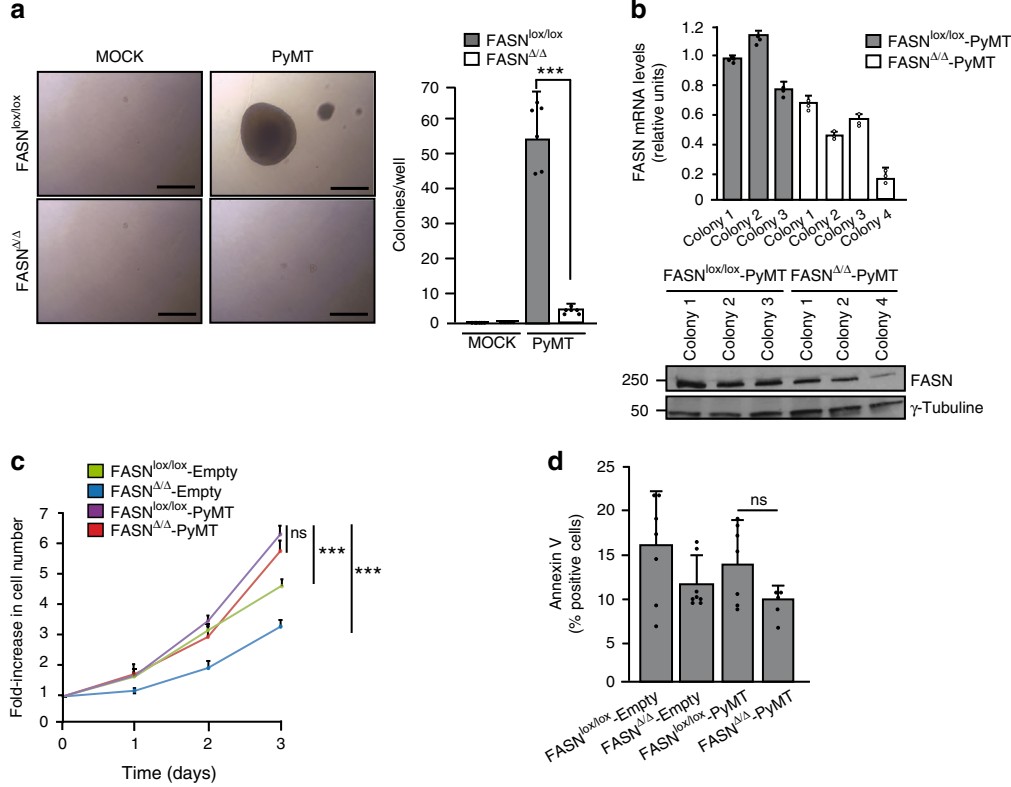

**Fig. 1** FASN is essential for PyMT-dependent transformation in vitro. **a** Representative images of the soft agar colony assay and quantitation of colonies per well recovered from each of the four genotypes ($n = 6$ biological replicates for each genotype). Scale bars, 500 μM. **b** FASN mRNA and protein levels detected in soft agar colonies recovered from the FASN$^{lox/lox}$-PyMT and FASN$^{\Delta/\Delta}$-PyMT MEFs, respectively. **c** Cell replication rates after 1, 2, or 3 days for each genotype ($n = 3$). Presented data are the mean values ± standard deviation (SD). ***$P < 0.001$; n.s., not statistically significant; Student's $t$ test. **d** Flow cytometry analysis of apoptosis. Percentages of cells in apoptosis are represented (FASN$^{lox/lox}$-Empty, $n = 7$; FASN$^{\Delta/\Delta}$-Empty, $n = 8$; FASN$^{lox/lox}$-PyMT, $n = 7$; FASN$^{\Delta/\Delta}$-PyMT, $n = 5$). Presented data are the mean values ± standard deviation (SD)

adding acetyl-CoA to FASN$^{lox/lox}$-PyMT lysates: pyruvate dehydrogenase activity was similar to that of FASN$^{\Delta/\Delta}$-PyMT MEFS, but not statistically significantly different—albeit lower— to that of FASN$^{lox/lox}$-PyMT MEFS (Supplementary Fig. 3a). FASN$^{lox/lox}$-PyMT MEFs showed the maximum glycolytic (reflected in extracellular acidification rate, ECAR) (Fig. 2c) and oxygen consumption rate (OCR) (Fig. 2d), matching other metabolic observations in transformed cells[31,50]. In the absence of the PyMT oncogene, both MEFs with and without FASN (FASN$^{lox/lox}$-Empty and FASN$^{\Delta/\Delta}$-Empty, respectively) showed similar ECAR and OCR. However, we observed a very low respiration (baseline and maximal respiratory capacity) and glycolytic capacity (similar to the non-transformed MEFs; Fig. 2c) when FASN was deleted prior to attempting transformation with PyMT (FASN$^{\Delta/\Delta}$-PyMT). The pre-transformation absence of FASN led to the same metabolic phenotype in HC-11 cells transformed with PyMT or KRAS (HC11-FASN$^{KO}$-PyMT and HC11-FASN$^{KO}$-KRAS, respectively) (Supplementary Fig. 3b) and in MEFs transformed with HER2 or KRAS (FASN$^{\Delta/\Delta}$-HER2 and FASN$^{\Delta/\Delta}$-KRAS, respectively) (Supplementary Fig. 3c). The lack of FASN also led to an inversion of the NAD$^+$/NADH quotient, indicating a decreased capacity of NAD$^+$ regeneration (Supplementary Fig. 3d).

Mitochondrial polarization is initiated by the generation of NADH/FADH during carbon skeleton flux along the Krebs cycle. ROS originate from electron dissipation that occurs when ATP is synthesized coupled to O$_2$ reduction by ATP synthase. The lack of negative mitochondrial charging (Fig. 2e) and the lack of ROS production in the FASN$^{\Delta/\Delta}$-PyMT MEFs (Fig. 2f) indicate that

the mitochondrial membrane is not polarized in the absence of FASN, explaining the low OCR observed.

Next, we performed carbon-tracing experiments with [U-$^{13}$C$_6$] glucose (Fig. 3a). Several findings are noteworthy: first, the levels of unlabeled lipids were similar in FASN$^{lox/lox}$-PyMT and FASN$^{\Delta/\Delta}$-PyMT MEFs; however, labeled lipid pools were virtually absent in the latter (Fig. 3b). This fact suggests the relative independence from FASN activity to maintain stable intracellular lipids levels and indicates that the abrogation of transformation is not dependent of FASN enzymatic product. Second, the comparative levels of certain glycolysis intermediate isotopomers (lactate CH$_3$ ions) and isotopologs [glucose-6-P (m + 6) and fructose-6-P (m + 6)] showed a >2-fold decrease in the glycolytic rate in FASN$^{\Delta/\Delta}$-PyMT MEFs compared with FASN$^{lox/lox}$-PyMT (Fig. 3c), regardless of the compensatory upregulation of Glut1 (trend) and Glut4 (statistically significant) in FASN$^{\Delta/\Delta}$-PyMT MEFs (Fig. 3d). Third, the rate of incorporation of carbon skeletons derived from glucose into the Krebs cycle (evidenced by the levels of the m + 2 isotopologs of citrate, aconitate, fumarate, and malate, and –CH$_2$ succinate isotopomer) was also diminished in FASN$^{\Delta/\Delta}$-PyMT MEFs compared with FASN$^{lox/lox}$-PyMT (Fig. 3e, f). Fourth, we observed a diminished incorporation of glucose-derived carbons of the m + 5 isotopoloe, corresponding to the ribose moiety of ADP ribose and AMP, in FASN$^{\Delta/\Delta}$-PyMT MEFs indicating a decrease in the pentose phosphate pathway flow (Fig. 3g). Taken together, these data confirm that when transformation is initiated in the absence of FASN, an accumulation of FASN substrates linked to a failure in the upregulation of glycolysis and the Krebs cycle is observed.

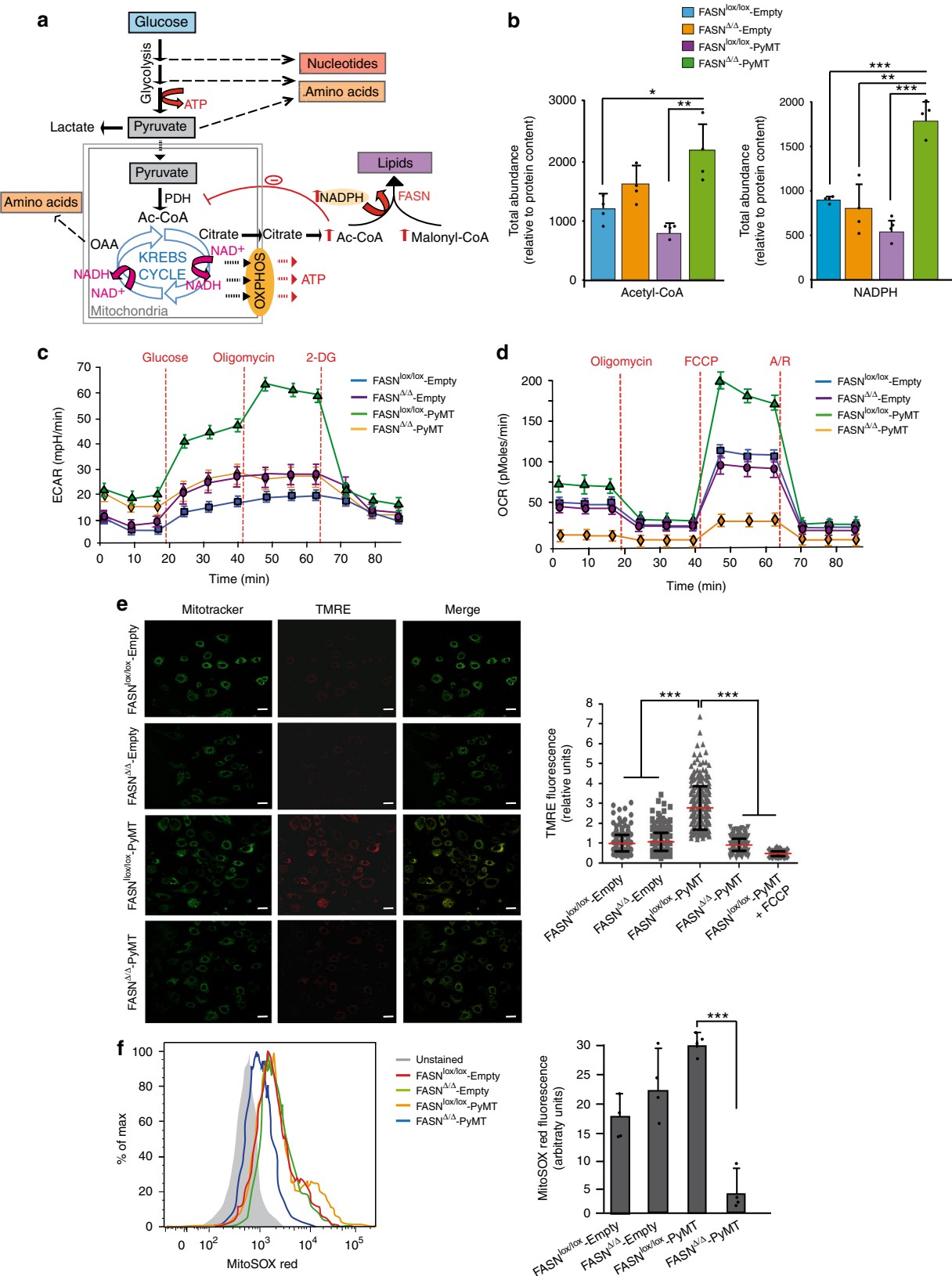

**Lack of FASN product does not explain the transform failure**. The observed metabolic phenotype could impair transformation for several reasons: (1) insufficient ATP production due to decreased glycolytic and/or respiratory rates[31]; (2) insufficient nucleotide production required for the DNA replication[31]; (3) although cancer cells can incorporate fatty acids from the surrounding media, the buildup of malonyl-CoA in the absence of FASN could block fatty acid oxidation (FAO)[51], a process shown to be essential in certain malignancies such as MYC-overexpressing breast cancer[52]; (4) the preferred source of palmitate for cancer cells could be the endogenously synthesized fatty acid by FASN. However, to our surprise, none of these four

**Fig. 2** FASN deficit impairs the upregulation of glycolysis and the anaplerotic shift of Krebs cycle. **a** The lack of FASN led to an accumulation of NADPH, acetyl-, and malonyl-CoA. The increase in acetyl-CoA can allosterically inhibit the activity of pyruvate dehydrogenase. This can block the entry of pyruvate into the Krebs cycle (associated with a decrease in the NAD-NADH cycling, respiration, and glycolytic rates) and the derivation of carbon skeletons to biosynthetic routes. In addition, a decrease in Krebs cycle activity could be coupled to a limited ability to satisfy high energetic demands. Pyruvate accumulation could also decrease glycolysis, which in addition to limiting ATP production would impair the ability to derive carbon skeletons to the nucleotide synthesis pathways. All these processes are upregulated in transformed compared with normal cells and are a hallmark of cancer[33]. **b** Total abundance of acetyl-CoA and NADPH determined by LC/MS across the four genotypes ($n = 4$). **c** Representative extracellular acidification rate (ECAR) measurements. Glucose, oligomycin, and 2-deoxyglucose (2-DG) were added at the indicated time point for each experiment. **d** Mitochondrial respiration reflected by oxygen consumption rate levels (OCR) was detected in the four genotypes. In **c**, **d** presented data are mean values ± SEM. **e** Mitotracker dye (green) was used to reveal the localization of mitochondria (left), and the positively charged dye TMRE (red) (middle) was used to compare mitochondrial transmembrane polarization. Representative fluorescence images of TMRE and Mitotracker were taken by confocal microscopy. Merged images are shown. Scale bars, 10 μM. Quantitative analysis of relative fluorescence intensity is shown in the right-hand side (FASN$^{lox/lox}$-Empty, $n = 388$ cells; FASN$^{\Delta/\Delta}$-Empty, $n = 458$ cells; FASN$^{lox/lox}$-PyMT, $n = 294$ cells; FASN$^{\Delta/\Delta}$-PyMT, $n = 382$ cells). FCCP was added to FASN$^{lox/lox}$-PyMT as a negative control ($n = 90$ cells). **f** Representative flow cytometry analysis (left) showing constitutive mitochondrial superoxide levels for each genotype ($n = 4$). The quantitation of MitoSOX red fluorescence intensity is shown in the right. In **b**, **e**, and **f** represented data are the mean values ± SD. *$P < 0.05$, **$P < 0.01$, and ***$P < 0.001$; Student's $t$ test

features found in the FASN$^{\Delta/\Delta}$-PyMT MEFs were definitively associated with the inability to transform. First, ATP levels and ADP/ATP quotients were not significantly different between FASN$^{lox/lox}$-PyMT and FASN$^{\Delta/\Delta}$-PyMT MEFs (Supplementary Fig. 4a). Second, we attempted to rescue the ability of PyMT to transform FASN$^{\Delta/\Delta}$ MEFs by adding dNTPs, and/or soraphen-A [an acetyl-CoA carboxylase inhibitor that decreases the availability of malonyl-CoA pools[53]], and/or palmitate to the media following the time points outlined in Supplementary Fig. 4b. Interestingly, in the presence of palmitate, neither soraphen-A nor dNTPs could rescue the transforming ability of PyMT in FASN$^{\Delta/\Delta}$ MEFs (Supplementary Fig. 4c). FAO of exogenous, [14]C-labeled fatty acids shows that both FASN$^{lox/lox}$-PyMT and FASN$^{\Delta/\Delta}$-PyMT MEFs oxidize fatty acids, although FASN$^{\Delta/\Delta}$-PyMT MEFs do so—as expected—at a lower rate (Supplementary Fig. 4d). The addition of soraphen-A, as predicted, boosted and rescued, respectively, FAO in FASN$^{lox/lox}$-PyMT and FASN$^{\Delta/\Delta}$-PyMT MEFs (Supplementary Fig. 4e), suggesting that although the malonyl-CoA buildup accounts for a decrease in FAO, this is not the reason why FASN$^{\Delta/\Delta}$-PyMT can not undergo transformation. Taken together, these observations indicate that although the lack of FASN affected several important cellular functions, they were not the ultimate cause of the failure to transform. In addition, this experiment showed that transformation was only possible in fatty acid-containing media (either free fatty acids (FFAs) or lipoproteins[36]), and that FASN$^{lox/lox}$-PyMT-HER2 or PyMT-KRAS MEFs could not form colonies in fatty acid-free media (Supplementary Fig. 4f). The role of exogenous fatty acid in colony formation essentiality was further tested: first, palmitate represents only 26% of fetal bovine serum (FBS) fatty acid pools[54]; thus, other exogenous lipids could be essential for transformation. When FASN$^{lox/lox}$-PyMT, FASN$^{lox/lox}$-HER2, or FASN$^{lox/lox}$-KRAS MEFs were incubated in full medium with FBS and the fatty acid uptake inhibitor sulfo-*N*-succinimidyl oleate (SSO)[55], no colonies were recovered. Second, when fatty acid-free media were supplemented with fatty acid-free albumin-complexed palmitate in order to facilitate its uptake by cancer cells, FASN$^{lox/lox}$-PyMT, FASN$^{lox/lox}$-HER2, or FASN$^{lox/lox}$-KRAS MEFs did not form colonies in soft agar (Supplementary Fig. 4f). Last, tumor epithelial cells from the breast of PyMT animals with (FASN$^{+/+}$; PyMT) or without (FASN$^{\Delta/\Delta}$; PyMT) FASN displayed similar intracellular neutral lipid accumulation evidenced by oil-red staining (Supplementary Fig. 4g). These results indicate that fatty acid requirements of cancer cells are mainly satisfied by the uptake from FFAs, implying that FASN is required for transformation for a different reason than its synthetic product.

**Upregulation of FASN elicits the 2D-to-3D switch.** Colony formation assays and tumor formation in vivo require 3D growth in the absence of extracellular matrix anchorage. Anchorage-independent growth is a hallmark of cancer[33]. Detachment from the matrix is associated with increased ROS production[34]. We hypothesized that the lack of tumorigenicity observed in the FASN-negative cells might be related to an insufficient capacity for quenching excessive ROS[32]. Therefore, we studied cell cultures under ultralow attachment conditions in which cancer cells form 3D tumor colonies or spheroids and mimic soft agar experiments, but allow easier experimental manipulation. In contrast with the 2D experiments in which FASN$^{\Delta/\Delta}$-PyMT and FASN$^{lox/lox}$-PyMT did not show significant differences in cell replication, the cell number and the size of spheroids recovered after 72 h of growth under ultralow attachment conditions of the FASN$^{\Delta/\Delta}$-PyMT MEFs were decreased in more than 75% compared with FASN$^{lox/lox}$-PyMT MEFs (Fig. 4a). Similar decreases in spheroid formation were observed in HC11 and MEFs expressing PyMT, KRAS, or HER2 when cells lacked FASN (Supplementary Fig. 4a). This phenotype was accompanied by a similar decrease in OCR as observed in 2D experiments (Fig. 4b). Regarding ROS levels, FASN$^{lox/lox}$-PyMT MEFs did not show significant changes in the transition from 2D to 3D growth. However, FASN$^{\Delta/\Delta}$-PyMT experienced a >20-fold increase in ROS in 3D compared with 2D conditions (Fig. 4c). Similar ROS increases were observed in 3D compared with 2D for the HER2, KRAS, and PyMT oncogenes in FASN-negative MEFs and HC11 cells (Supplementary Fig. 4b).

Reportedly, ROS levels must be maintained within certain boundaries to allow tumors to develop[32]. Thus, an excess of ROS could account for the inability to transform in the absence of FASN. In that case, forcing ROS production in FASN$^{lox/lox}$-PyMT MEFs should phenocopy FASN deletion. We treated FASN$^{lox/lox}$-PyMT MEFs cultured in 3D with 10 μM DMNQ (a cell-permeable redox cycling quinone that produces intracellular superoxide anion). At this dose level, DMNQ produced similar ROS levels to those observed in FASN$^{\Delta/\Delta}$-PyMT MEFs in 3D (Fig. 4d). As expected, DMNQ disrupted spheroid formation (Fig. 4e).

A recent report showed that in 2D cancer cultures, glutamine is transformed into α-ketoglutarate via glutamate and then oxidized along the Krebs cycle; however, in 3D cultures, glutamine-derived α-ketoglutarate was produced in the cytosol and transformed into isocitrate/citrate via IDH1-dependent reductive carboxylation[39,40]. This pair of metabolites can be shuttled back into the mitochondria and enter the Krebs cycle generating intra-mitochondrial NADPH to quench the 3D ROS excess by

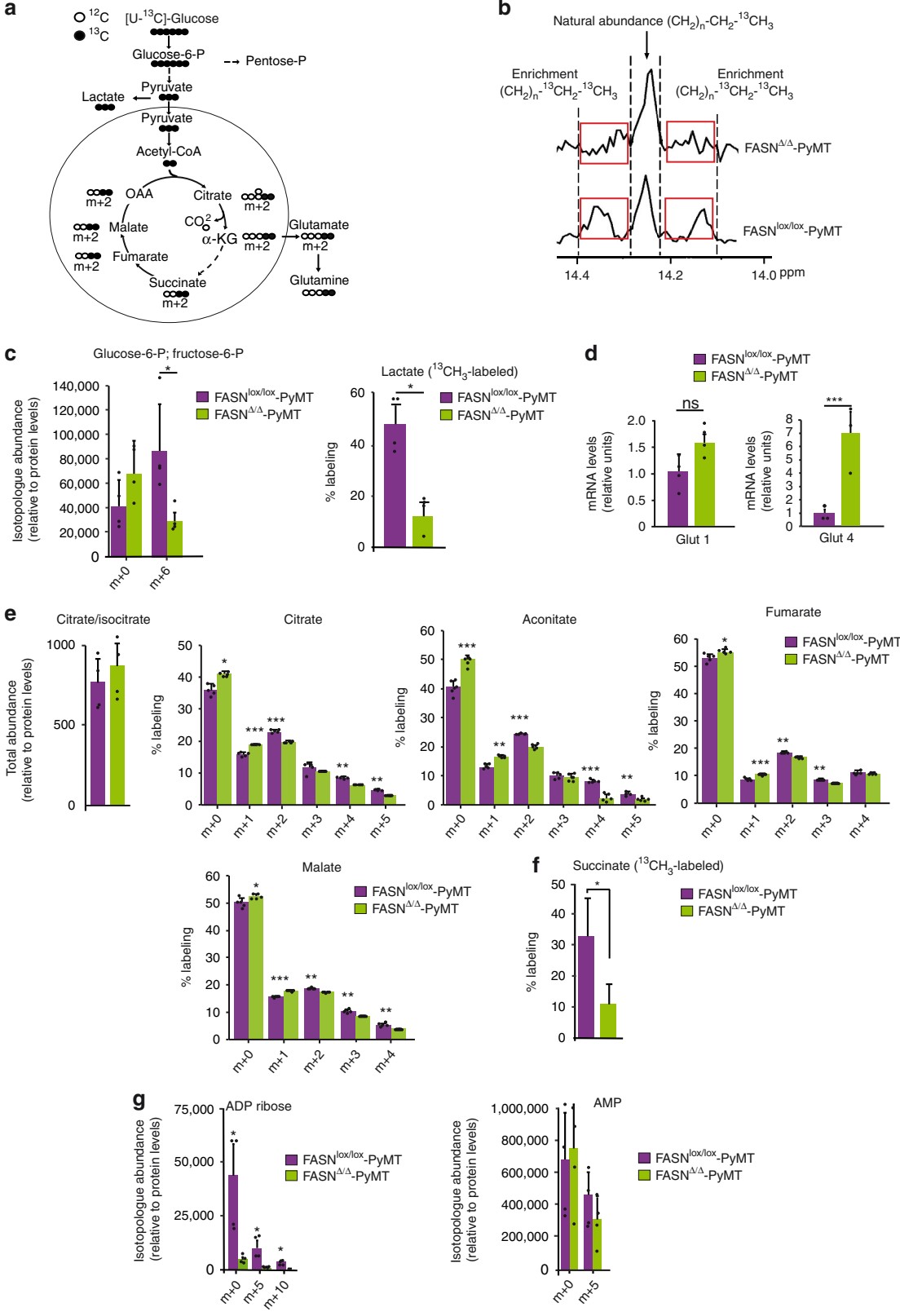

maintaining glutathione (GSH) levels[40]. Among the potential effects derived from the lack of FASN catalytic activity, acetyl-CoA accumulation could block the activity of ATP citrate lyase, leading to a buildup of cytoplasmic citrate/isocitrate that would stall the

IDH1-dependent reductive carboxylation required for sustaining 3D growth. Although the citrate/isocitrate m + 2 was reduced in FASN$^{\Delta/\Delta}$-PyMT (Fig. 3e), reflecting a decrease in pyruvate dehydrogenase activity, the total citrate/isocitrate levels were

**Fig. 3** Decreased glucose carbon flux and the pentose phosphate pathway. **a** Schematic representing incorporation of $^{13}$C derived from glucose into metabolites of glycolysis and the Krebs cycle. White and black circles are $^{12}$C and $^{13}$C, respectively. **b** Decreased labeled but similar total palmitate pools in FASN$^{\Delta/\Delta}$-PyMT MEFs compared with FASN$^{lox/lox}$-PyMT MEFs. $^{13}$C-NMR spectrum showing the central peak (singlet) at 14.17 p.p.m. characteristic of the terminal methyl group (–CH$_3$ or omega-1 carbon) of a fatty acid, and lateral peaks (doublet signal) corresponding to the penultimate –CH$_2$- (or omega-2 carbon) due to $^{13}$C–$^{13}$C coupling. **c** Isotopolog abundance of glucose-6-P (determined by LC/MS) and lactate (determined by RMN). **d** Glut1 and Glut4 glucose transporter mRNA levels determined by RT-qPCR, relativized to FASN$^{lox/lox}$-PyMT values. The experiment was repeated three times. **e** Total abundance of citrate/isocitrate and % labeling of citrate, aconitate, fumarate, and malate isotopologs in MEFs labeled with [U-$^{13}$C]glucose (determined by LC/MS) ($n = 5$ biological replicates), and **f** % labeling of –CH$_2$ succinate isotopomer (determined by RMN) ($n = 5$ biological replicates). **g** Isotopolog abundance of ADP ribose and AMP determined by LC/MS in FASN$^{lox/lox}$-PyMT and FASN$^{\Delta/\Delta}$-PyMT MEFs ($n = 5$ biological replicates). Presented data are the mean values ± SD. *$P < 0.05$, **$P < 0.01$, and ***$P < 0.001$; n.s., not statistically significant; Student's $t$ test

unchanged (Fig. 3e). Concordantly, both the addition of sodium citrate or SB204990 (an ATP-citrate lyase inhibitor) to the 3D cultures of FASN$^{lox/lox}$-PyMT phenocopied FASN$^{\Delta/\Delta}$-PyMT MEFs: (1) no spheroids were recovered (Fig. 4f); (2) ROS levels were similar to those observed in FASN$^{\Delta/\Delta}$-PyMT MEFs (Fig. 4f); (3) OCR was suppressed (Fig. 4g). As expected, ATP citrate lyase activity was significantly decreased in FASN$^{\Delta/\Delta}$-PyMT MEFs compared with FASN$^{lox/lox}$-PyMT MEFs (Fig. 4h). Citrate and SB204990 also disrupted spheroid formation and decreased OCR/ECAR in FASN$^{lox/lox}$ MEFs or HC11 cells transformed with KRAS, HER2, or PyMT (Supplementary Fig. 5a, b). In 2D growth, MEF and HC11 cell viability was unaffected by the addition of citrate or SB204990 at the concentration used in previous experiments (5 mM and 30 μM in MEFs; 3 mM and 30 μM in HC11, respectively; Fig. 4i, j, Supplementary Fig. 6c, d). When IDH1 was silenced with small-interfering RNA (siRNA) (Supplementary Fig. 10a) in 3D, it led to a disruption in the formation of tumor spheroids (Supplementary Fig. 10b) and suppression of OCR (Supplementary Fig. 10c) together with increased ROS (Supplementary Fig. 10d), compared with no effect in these traits in 2D. Taken together, these data suggest that in the absence of FASN, citrate lyase activity is diminished, and this enzymatic impairment is linked to a defect in 3D, but not 2D, tumor cell growth.

Carbon-tracing experiments with [U-$^{13}$C$_5$]glutamine demonstrated that in the absence of FASN, cells showed a decreased ability to perform reductive carboxylation; however this limitation was restricted to 3D growth (Fig. 5d–f) and not observed in 2D growth (Fig. 5a–c). Increased α-ketoglutarate m + 5 and fumarate m + 4 labeling from [U-$^{13}$C$_5$]glutamine in FASN$^{\Delta/\Delta}$-PyMT MEFs under 2D conditions revealed that in the absence of FASN, MEFs may compensate the decreased entry of carbon skeletons from glucose into the Krebs cycle by enhancing the oxidative metabolism of glutamine (Fig. 5a, b). In addition, no changes in citrate/isocitrate m + 5 levels were observed between genotypes, indicating that glutamine enters the mitochondria and participates predominately in oxidative metabolism (Fig. 5c). However, in 3D, isotope tracing revealed that the reductive formation of citrate/isocitrate m + 5 from glutamine was enhanced in FASN$^{lox/lox}$-PyMT MEFs. This was observed only in FASN$^{lox/lox}$-PyMT MEFs, indicating active IDH1-dependent reductive carboxylation (Fig. 5d, e, f). These data suggest that IDH1-dependent reductive carboxylation is halted in FASN$^{\Delta/\Delta}$-PyMT MEFs. In this situation, theoretically, there would be insufficient intra-mitochondrial reduced equivalents, which would be consumed by the excess of unquenched ROS produced during the 2D–3D transition. This would be followed by an impairment of mitochondrial respiration that should be rescued by the addition of reductive equivalents. To test this hypothesis, first we measured intra-mitochondrial and total NADPH. Total NADPH levels were decreased in FASN$^{\Delta/\Delta}$-PyMT compared with FASN$^{lox/lox}$-PyMT MEFs (Fig. 5g). However, the difference was more pronounced in the intra-mitochondrial compartment

(Fig. 5g), which is congruent with the fact that cytoplasmic levels may still be high because of the lack of activity of FASN, which consumes NADPH. It has been shown that an increased ROS production could disrupt the assembly of mitochondrial complexes into supercomplexes[56,57], stalling respiration. While in 2D the decreased respiration levels of FASN$^{\Delta/\Delta}$-PyMT can be attributed to a decreased entrance of pyruvate in the mitochondria, in 3D the increased ROS might be a more important contributor. In fact, we observed that in 2D total NADPH was increased in FASN$^{\Delta/\Delta}$-PyMT clones (Fig. 2b); however, in 3D in the context of decreased citrate lyase activity, we observed decreased intramitochondrial NADPH levels (Fig. 5g), increased ROS (Fig. 4c), and a decreased proportion of complex I assembly into supercomplexes (Fig. 5h). In order to ascertain the functional impact of these observations, we cultured FASN$^{\Delta/\Delta}$-PyMT MEFs in the presence of either N-acetyl-cysteine (NAC) or glutathione monoethyl ester (GSH-MEE), two ROS-quenching agents. This approach rescued OCR (Fig. 5i), tumor spheroid formation, and soft agar colony formation (Fig. 5j) while reducing ROS burden in 3D (Fig. 5k). The NAD$^+$/NADH quotient was also restored after the addition of NAC/GSH-MEE (Fig. 5l). We observed similar rescues in the other studied FASN-negative systems expressing different oncogenes (Supplementary Fig. 6a, b). Taken together, our data suggest that the acetyl-CoA buildup secondary to FASN deletion inhibits ATP citrate lyase and induces an accumulation of citrate/isocitrate. Subsequently, this impairs IDH1-dependent reductive carboxylation, a tumor cell requirement limited to 3D growth, inducing a decrease in intramitochondrial NADPH, an increase in ROS, and finally mitochondrial supercomplexes' assembly disruption, which ultimately impairs cell transformation and tumorigenesis.

**FASN deletion prevents formation of invasive tumors.** In vivo, the PyMT oncogene expressed under the control of the MMTV promoter in a pure FVB background generates invasive ductal carcinoma of the breast that recapitulates human "luminal-B" tumors in ~100% of the animals at the age of 7 weeks[58]. Figure 6a shows FASN staining in breast sections of wild-type and MMTV-PyMT animals at 4, 8, and 13 weeks of age. This time course shows that FASN upregulation is associated with tumor development. We crossed FASN$^{lox/lox}$;Tg UBC-CreER$^{T2}$ mice with MMTV-PyMT animals and induced systemic Cre activity with tamoxifen when animals were weaned off (4–5 weeks of age). Tamoxifen was also administered to the control group. Importantly, FASN$^{+/+}$; PyMT and FASN$^{\Delta/\Delta}$; PyMT animals showed similar fasted triglyceride blood levels (Supplementary Fig. 7a). The PyMT is an extraordinarily penetrant oncogene (hundreds of tumor foci are simultaneously developed in each mammary gland), and Cre recombinase activity was not sufficient to delete the FASN gene in 100% of the breast epithelial cells (Fig. 6b), thus allowing the development of FASN-positive tumor foci. However, FASN-negative regions did not develop breast tumors (Fig. 6b). Based on our previous in vitro data, an insufficient capacity for

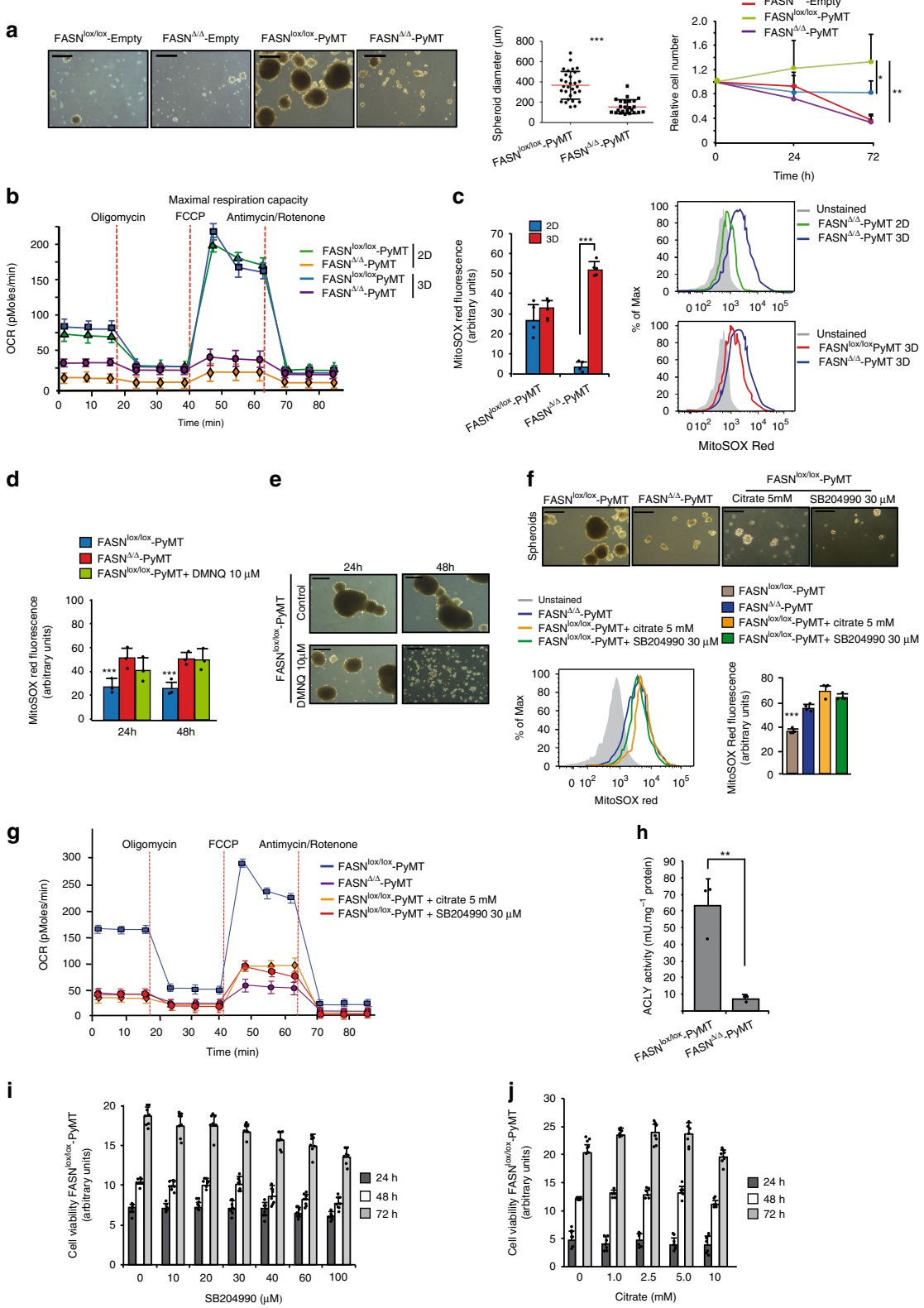

quenching excess ROS might be related to the lack of breast tumor development in FASN-negative regions in PyMT animals. By using an oxidative damage DNA biomarker, anti-8-oxo-dG that specifically binds to 8-oxo-2′-deoxyguanosine (8-oxo-dG), one of the major products of DNA oxidation, we showed an increase in ROS levels in FASN$^{\Delta/\Delta}$; PyMT tumors (Fig. 6c).

Interestingly, this ROS accumulation in vivo corresponds to FASN-negative areas evidenced by the detection of FASN and 8-oxo-dG co-staining with monoclonal antibodies (Abs)[59] (Fig. 6d). The Kaplan–Meier curve shown in Fig. 6e depicts the differences in time to killing between FASN$^{+/+}$; PyMT and FASN$^{\Delta/\Delta}$; PyMT animals. Tumor volume is depicted in Fig. 6f. FASN$^{\Delta/\Delta}$; PyMT

**Fig. 4** FASN deletion leads to ATP citrate lyase (ACLY) inhibition, blocking 3D growth. **a** Representative pictures of the spheroids recovered from the different genotypes growing under ultralow attachment conditions for 72 h (left). Spheroid diameter ($n = 30$ spheroids, FASN$^{lox/lox}$-PyMT; $n = 24$ FASN$^{\Delta/\Delta}$-PyMT) and number of cells over 72 h (right). Scale bars, 500 μM. Presented data are the mean values ± SD. ***$P < 0.001$; Student's $t$ test. **b** OCR assay performed with cells recovered from the previous experiment and monolayer cultures. Oligomycin, FCCP, and antimycin/rotenone were added at the indicated time point. Data are represented as the means ± SEM. **c** Representative flow cytometry analysis showing constitutive mitochondrial superoxide levels in monolayer compared with 3D conditions in PyMT MEFs loaded with MitoSOX Red (FASN$^{lox/lox}$-PyMT, $n = 4$; FASN$^{\Delta/\Delta}$-PyMT, $n = 5$). The quantification of MitoSOX Red fluorescence intensity is shown on the left. Presented data are the mean values ± SD. ***$P < 0.001$, Student's $t$ test. **d** Quantification of MitoSOX Red fluorescence intensity by flow cytometry analysis in FASN$^{lox/lox}$-PyMT, FASN$^{\Delta/\Delta}$-PyMT, and DMNQ-treated (10 μM; 20 min) FASN$^{lox/lox}$-PyMT MEFs ($n = 3$). **e** Effect of 10 μM DMNQ treatment in the ability of FASN$^{lox/lox}$-PyMT MEFs to form spheroids. Scale bars, 500 μM. Representative pictures for each condition are shown. **f** Representative pictures showing the effects of sodium citrate (5 mM) or SB204990 (30 μM) in the formation of tumor spheroids by FASN$^{lox/lox}$-PyMT MEFs. Scale bars, 500 μM. The lower charts show an increase in ROS production by sodium citrate and SB204990. Representative flow cytometry analysis was shown ($n = 4$). **g** Effects in OCR suppression caused by sodium citrate or SB204990 treatment in FASN$^{lox/lox}$-PyMT MEFs. Presented data are the mean values ± SEM. Three independent experiments (**h**) ACLY activity in FASN$^{lox/lox}$-PyMT versus FASN$^{\Delta/\Delta}$-PyMT MEFs. **i, j** Cell viability was determined in FASN$^{lox/lox}$-PyMT MEFs at 24, 48, and 72 h in the presence of various concentrations of SB204990 ($n = 8$) (**i**) or sodium citrate ($n = 7$) (**j**) under 2D culture conditions. In **h**–**j** represented data are the mean values ± SD. *$P < 0.05$, **$P < 0.01$, and ***$P < 0.001$; Student's $t$ test

animals ultimately also die due to the incomplete deletion of FASN in the mammary tissue and development of PyMT/FASN-positive tumors (although the number of tumor foci is smaller in FASN-negative tumors, a single focus is sufficient to generate a tumor that will mandate animal killing; Fig. 6b, arrows). In order to achieve complete abrogation of FASN activity, we combined genetic plus pharmacologic approaches. G28UCM is a compound with FASN inhibitory properties[27]. Although this compound is not clinical grade, it has been used to inhibit FASN effectively in preclinical experiments. In vitro, G28UCM recapitulated the main features of FASN deletion and showed moderate on-target effect (Supplementary Fig. 8a–d). In vivo, when applied to FASN$^{\Delta/\Delta}$; PyMT animals, G28UCM induced further tumor growth arrest; however, the combination was quite toxic and the experiment had to be interrupted after 3 weeks. It has been shown that FASN deletion in the intestinal epithelium can be lethal[18]. FASN deletion by using UBC-CreER$^{T2}$ was incomplete and the animals tolerated it; however, in combination with G28UCM, the animals suffered from intense diarrhea and weight loss akin the intestinal FASN deletion study; Supplementary Fig. 8e. In addition, in agreement with the proposed mechanism of action of FASN deletion, when G28UCM was applied to FASN$^{lox/lox}$; PyMT animals, their tumors displayed intense 8-oxo-dG staining (Supplementary Fig. 9).

We also tested the in vivo effects on tumor growth of FASN$^{\Delta/\Delta}$ MEFs infected with KRAS or HER2, grafting them into wild-type animals compared with FASN$^{lox/lox}$ counterparts (Supplementary Fig. 7b, c). In line with the in vitro data (Supplementary Fig. 2g), we did not observe different replication levels (Ki67 staining) (Supplementary Fig. 7f) or oncogene levels (KRAS, HER2) in tumors originated from FASN$^{lox/lox}$ or FASN$^{\Delta/\Delta}$ clones (Supplementary Fig. 7g, h). A strong reduction in tumor growth was observed. Finally, when HC11 cells with CRISPR-deleted FASN were infected with PyMT or KRAS and grafted into wild-type animals, a delay on tumor onset was observed compared with wild-type counterparts (Supplementary Fig. 7d, e). Importantly, the positive antitumor effects observed in these xenografts were produced in a FASN wild-type background in the rest of the organism, supporting the concept that the effects observed in the PyMT model were independent of systemic FASN deletion.

Taken together, in vivo and in vitro data suggest that FASN is essential for cell transformation and tumor development. We investigated a potential cancer prevention role of FASN inhibitors. Based on our data, this approach should be effective at the preventive stage. Continuous G28UCM treatment administration from 5 weeks of age in MMTV-PyMT animals significantly delayed tumor onset and growth (Fig. 7a).

Concordantly, G28UCM treatment also had an impact on the overall survival of mice (Fig. 7b). Animals did not lose weight during treatment (Fig. 7c). Although animals treated with G28UCM experienced tumor development, histological analysis showed that these tumors did not acquire the invasive, malignant phenotype, and growth was limited to in situ ductal carcinoma. The immunohistochemical staining shown in Fig. 7d depicts how FASN staining was intense in both vehicle- and G28UCM-treated tumors; however, collagen IV staining (a staining against the most abundant collagen molecule present in the basement membrane of breast gland ducts and used in the clinical setting to distinguish invasive from in situ carcinomas) showed that although epithelial cells proliferated and obliterated the ducts in the latter, the basement membrane preserved its integrity in contrast with the architecture-disrupting pattern observed in vehicle-untreated animals. Based on these results, FASN inhibitors show potential for tumor prevention when a clinical-grade compound becomes available.

## Discussion

FASN has been extensively studied in cancer, but its specific mechanistic relationship with carcinogenesis, and its definitive therapeutic role, have not been completely established. In the present study, the role of FASN during transformation was investigated and several significant results were found.

First, conceptually, disrupting metabolic events essential for transformation could be more critical in clinical care (cancer prevention) than after the cancer has already developed (cancer therapeutics) when it can rely on different backup processes, and has already overcome several rate-limiting steps of tumorigenesis. Based on our data, FASN is essential for PyMT-mediated transformation. MEF data, where the efficiency of adeno-Cre was >95% and all the colonies recovered from the FASN-floxed clones were the result of the residual adenoviral infection inefficiency (Fig. 1b), clearly support this statement. The regional distribution of FASN deletion achieved in vivo also supports that in the absence of FASN, the breast epithelium cannot undergo transformation (Fig. 6b). A recently developed acetyl-CoA carboxylase inhibitor[60] that blocks the generation of malonyl-CoA, a rate-limiting step of fatty acid synthesis, has showed impressive antitumor effects; several FASN inhibitors have achieved similar results, although most compounds have significant off-target effects[26,27,29,30]. However, antitumor efficacy has proven to be always transient and tumors always regrow; thus, curing tumors by using this strategy seems unlikely. From the clinical point of view, the effects of targeting FASN before rather than after transformation can only be emphasized.

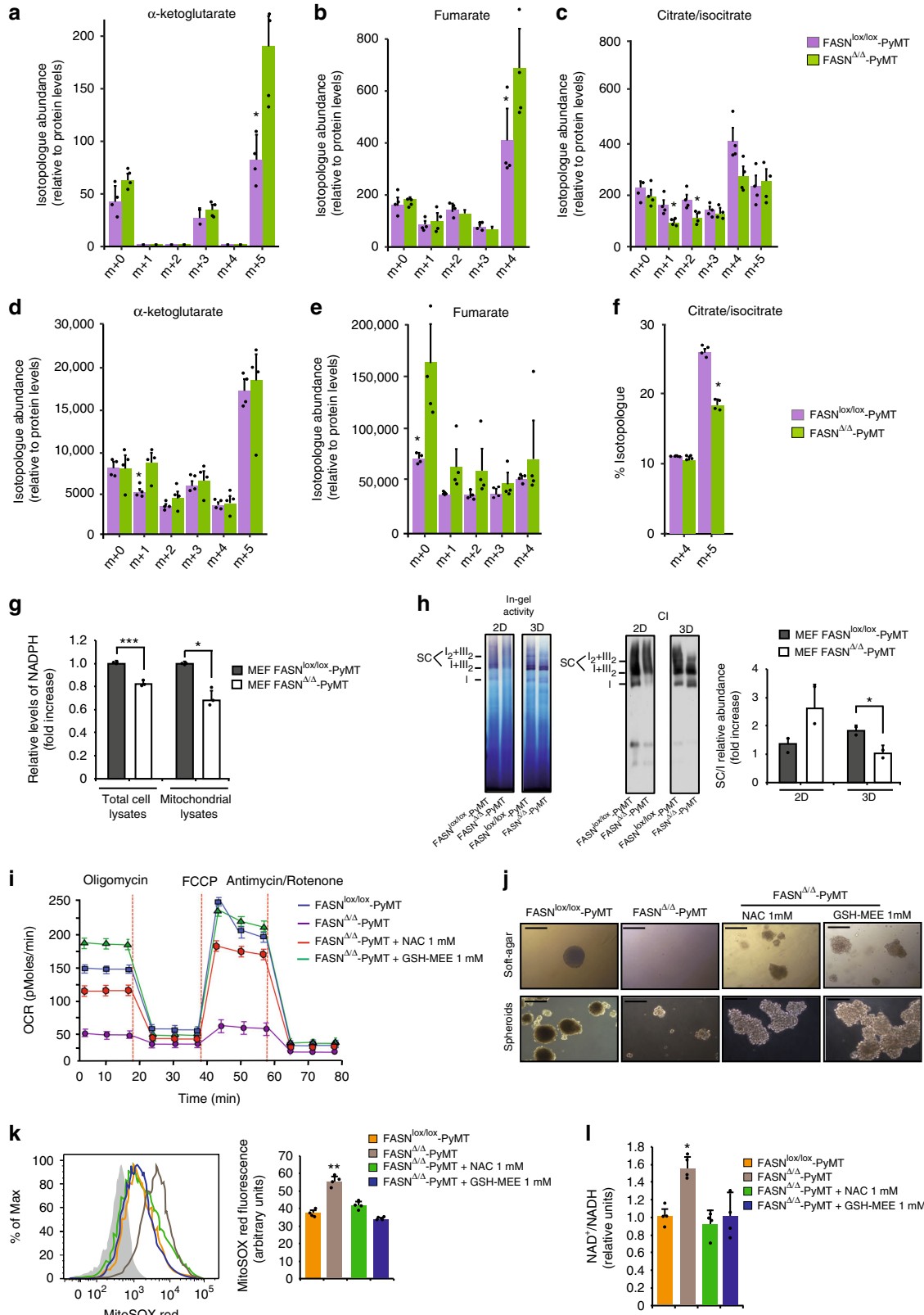

A very important finding from this study is that the importance of FASN for cancer development is independent of its biosynthetic product. Previous studies relied on the assumption that blocking FASN activity would limit the fatty acid pools available for building different cellular membranes required for proliferation[1,37,38]. While this effect may be important, our data

showed that in the pre-transformation state, re-supplementation with palmitate and fatty acids did not rescue the phenotype caused by FASN loss, indicating a different implication of FASN in the carcinogenesis process. Adult mammals rely mostly on dietary fatty acids to maintain peripheral fatty acid pools. Therefore, the unchanged levels of blood fatty acids observed in

**Fig. 5** FASN deficit leads to an impairment of IDH1-dependent reductive carboxylation. **a** Isotopolog abundance of $\alpha$-ketoglutarate, **b** fumarate, and **c** and citrate/isocitrate from [U-$^{13}$C$_5$]glutamine in 2D cultures determined by LC/MS ($n = 4$ biological replicates). **d** Isotopolog abundance of $\alpha$-ketoglutarate, **e** fumarate, and **f** citrate/isocitrate from [U-$^{13}$C$_5$]glutamine in 3D cultures determined by LC/MS ($n = 4$ biological replicates). Presented data are the mean values ± SD. *$P < 0.05$; Student's $t$ test. **g** Relative levels of NADPH in cellular and mitochondrial lysates of FASN$^{lox/lox}$-PyMT versus FASN$^{\Delta/\Delta}$-PyMT MEFs in 3D cultures ($n = 3$). Presented data are the mean values ± SD. *$P < 0.05$ and ***$P < 0.001$; Student's $t$ test. **h** Digitonin-solubilized isolated mitochondria from FASN$^{lox/lox}$-PyMT and FASN$^{\Delta/\Delta}$-PyMT MEFs in 2D and 3D culture conditions were subjected to blue native gel electrophoresis (BNGE) followed by in-gel complex I activity assay. Complex I occurs both free and bound with complex III (I + III$_2$ and I$_2$ + III$_2$ supercomplexes). Relative abundance of assembled complex I into supercomplexes (SC) versus free complex I (CI) after the quantification of band intensity in direct immunoblotting of BNGEC against complex I subunit NDUFS1 in FASN$^{lox/lox}$-PyMT versus FASN$^{\Delta/\Delta}$-PyMT in 2D and 3D culture conditions is shown. Presented data are the mean values ± SD ($n = 2$ independent culture preparation). *$P < 0.05$; Student's $t$ test. **i** NAC (1 mM) and GSH-MEE (1 mM) rescue experiments. Data are represented as mean values ± SEM. Three independent experiments. **j** Rescue of soft-agar colonies (above) and spheroids (below) in FASN$^{\Delta/\Delta}$-PyMT MEFs treated with NAC (1 mM) or GSH-MEE (1 mM). Representative images for each condition are shown. Scale bars, 500 μM. **k** Representative flow cytometry analysis showing constitutive mitochondrial superoxide levels in FASN$^{\Delta/\Delta}$-PyMT MEFs treated with NAC or GSH-MEE ($n = 4$). MitoSOX red fluorescence intensity quantitation is shown on the right. Presented data are the mean values ± SD. **$P < 0.01$; Student's $t$ test. **l** NAD$^+$/NADH ratio is restored after the addition of NAC or GSH-MEE in FASN$^{\Delta/\Delta}$-PyMT MEFs in 3D cultures ($n = 4$). Presented data are the mean values ± SD. *$P < 0.05$; Student's $t$ test

the FASN$^{\Delta/\Delta}$; PyMT mice compared with FASN$^{+/+}$; PyMT mice (Supplementary Fig. 7a) and the absence of tumor formation in wild-type animals grafted with FASN-negative KRAS-, HER2-, or PyMT-expressing cells (Supplementary Fig. 7b–e) show that the observed abrogation of breast epithelium transformation is not secondary to systemic effects of FASN deletion, and the availability in blood of the product that FASN synthesizes does not rescue the transforming phenotype. The product-independent role of FASN is further supported by the following facts: (1) in 2D conditions, FASN$^{\Delta/\Delta}$-PyMT MEFs grow normally (Fig. 1c) but not in 3D conditions (Fig. 4a); (2) palmitate or FFA-free albumin-complexed palmitate do not rescue tumorigenesis in FASN$^{\Delta/\Delta}$-PyMT MEFs (Supplementary Fig. 4c, f); (3) the preferential source of fatty acids appears external, and maybe relying on fatty acids different from palmitate, evidenced by FASN$^{lox/lox}$-PyMT MEFs with intact FASN levels forming colonies only in the presence of fatty acids in the media, but losing the ability to form colonies in the presence of SSO (Supplementary Fig. 4f). Data from the measurement of unlabeled lipid pools in both genotypes (Fig. 3b) and neutral lipid accumulation in the tumor epithelium of PyMT animals (Supplementary Fig. 4g) further support this conclusion. Although the lack of FASN affected several other metabolic pathways associated with transformation[31] (Fig. 2a), it did not account for the inability to transform cells as evidenced by the reconstitution experiments.

Finally, probably the most relevant finding is the role of FASN facilitating the transition from 2D to 3D growth. In normal cells, the main function of FASN is long-chain fatty acid synthesis; however, transforming cells can harness this enzymatic reaction to support transformation by eliciting the acquisition of a hallmark of cancer, 3D anchorage-independent growth[33]. In the absence of FASN, the production of reduced equivalents from IDH1-dependent reductive carboxylation to quench the excessive ROS observed in the 2D–3D transition is impaired, which leads to the observed decrease in intra-mitochondrial NADPH (Fig. 5g), and decreased levels of mitochondrial complex I assembly into supercomplexes (Fig. 5h), which accounts for the inability to transform (Figs. 4 and 5). This was proven by carbon tracing, enzymatic activity determination, ROS levels, mitochondrial complex integrity analysis, and phenocopying/phenotype-rescuing experiments with several metabolites and inhibitors across several different genotypes. Excessive ROS were observed in FASN$^{\Delta/\Delta}$; PyMT animals in the FASN-negative PyMT-positive areas that proliferate but did not acquire invasive growth properties (Fig. 6d). Similarly, when G28UCM was administered in vivo, invasive growth was not observed (Fig. 7d). Based on our data, FASN would be a necessary checkpoint for the acquisition of this essential hallmark of cancer.

FASN has been previously related to relevant diverse features of tumor progression such as increased cell replication[1] or HER2 signaling[61]. In the present study, we described a mechanism regarding the role of FASN in cancer, linking the very common upregulation of FASN in cancer to a critical role in acquiring 3D growth properties during transformation, unrelated to its biosynthetic product. Because endogenous FASN is not required in most adult tissues and due to the specificity of the IDH1-dependent reductive carboxylation process in cancer cells, this mechanism is a highly attractive, cancer-specific target. With novel FASN inhibitors in perspective[30,62,63], a clinical-grade compound that would selectively target FASN could be administered long term to a healthy individual, by sparing toxicity to normal tissues, while it could target cells in their initial steps of transformation. Taken together, these features indicate that FASN is a potential target for cancer prevention. Future studies with clinical-grade compounds in high-risk patient subpopulations (akin previous chemoprevention trials in breast cancer[64,65]) could address the therapeutic utility of this strategy.

## Methods

**Generation of FASN-knockout mice**. FASN$^{lox/lox}$ mice were generated from knockout embryonic stem (ES) cells for *FASN* (C57BL/6N-*FASN* tm1(KOMP)wtsi) obtained from the KOMP Repository (www.komp.org) (KOMP Repository, UC Davies)[66]. ES cells were microinjected into C57BL/6 mouse blastocysts, from which chimeric males were obtained. Chimeras were bred with C57BL/6 wild-type females to produce heterozygous floxed (FASN$^{flox/+}$) mice.

After successful germ line transmission, FASN$^{flox/+}$ mice were crossed with Flp transgenic mice to facilitate in vivo frt-neo deletion (FASN$^{lox/+}$). To obtain FASN$^{lox/+}$ mice with the FVB background, we backcrossed FASN$^{lox/+}$ heterozygous males with FVB females by using marker-assisted selection protocol (Speed Congenics). At every generation, the mouse carrying the fewest C57BL/6 background loci was used for backcrossing. After six generations, a close to 100% of FVB background was achieved. Then, FASN$^{lox/lox}$ mice were bred with Cre recombinase-expressing transgenic mice to achieve deletion of the targeted region. Tg-UBC-CreER$^{T24i}$ were obtained from the CNIO Animal House (Madrid, Spain). PyMT [FVB/N-Tg (MMTV-PyVT$^{634Mul/J}$)] mice were obtained from W. Muller (McMaster University, Ontario, Canada). All mice were on a FVB pure background with littermate controls (females) used in all experiments. Genotyping was performed by PCR analysis of tail DNA. Primers used for genotyping FASN wild-type (wt) and floxed mice were common forward (5′-ATGGATTACCCAAGCGGTCT-3′) and common reverse (5′-CCTGTCTCTGAGCCCTTGAT-3′). Polymerase chain reaction conditions were as follows: 95 °C for 15 min; 94 °C for 30 s; 35 cycles at 57 °C for 30 s; 72 °C for 30 s; 72 °C for 10 min; then soaking at 4 °C. Primers used for genotyping Cre allele were forward—5′-GCTCGACCAGTTTAGTTACCC-3′ and reverse—5′-TCGCGATTATCTTCTATATCTTCAG-3′. Polymerase chain reaction conditions were as follows: 95 °C for 15 min; 94 °C for 30 s; 35 cycles at 64 °C for 30 s; 72 °C for 30 s; 72 °C for 10 min; then soaking at 4 °C. Primers used for genotyping *PyMT* transgene were 5′-GGAAGCAAGTACTTCACAAGG-3′ and 3′-GGAAA GTCACTAGGAGCAGGG-5′. Polymerase chain reaction conditions were as follows: 95 °C for 15 min; 94 °C for 30 s; 30 cycles at 59 °C for 45 s; 72 °C for 1 min; 72 °C for 10 min; then soaking at 4 °C. PCR products are 336 bp (base pair) for wt allele, 438 bp for lox allele, 470 bp for cre allele, and 557 bp for PyMT allele.

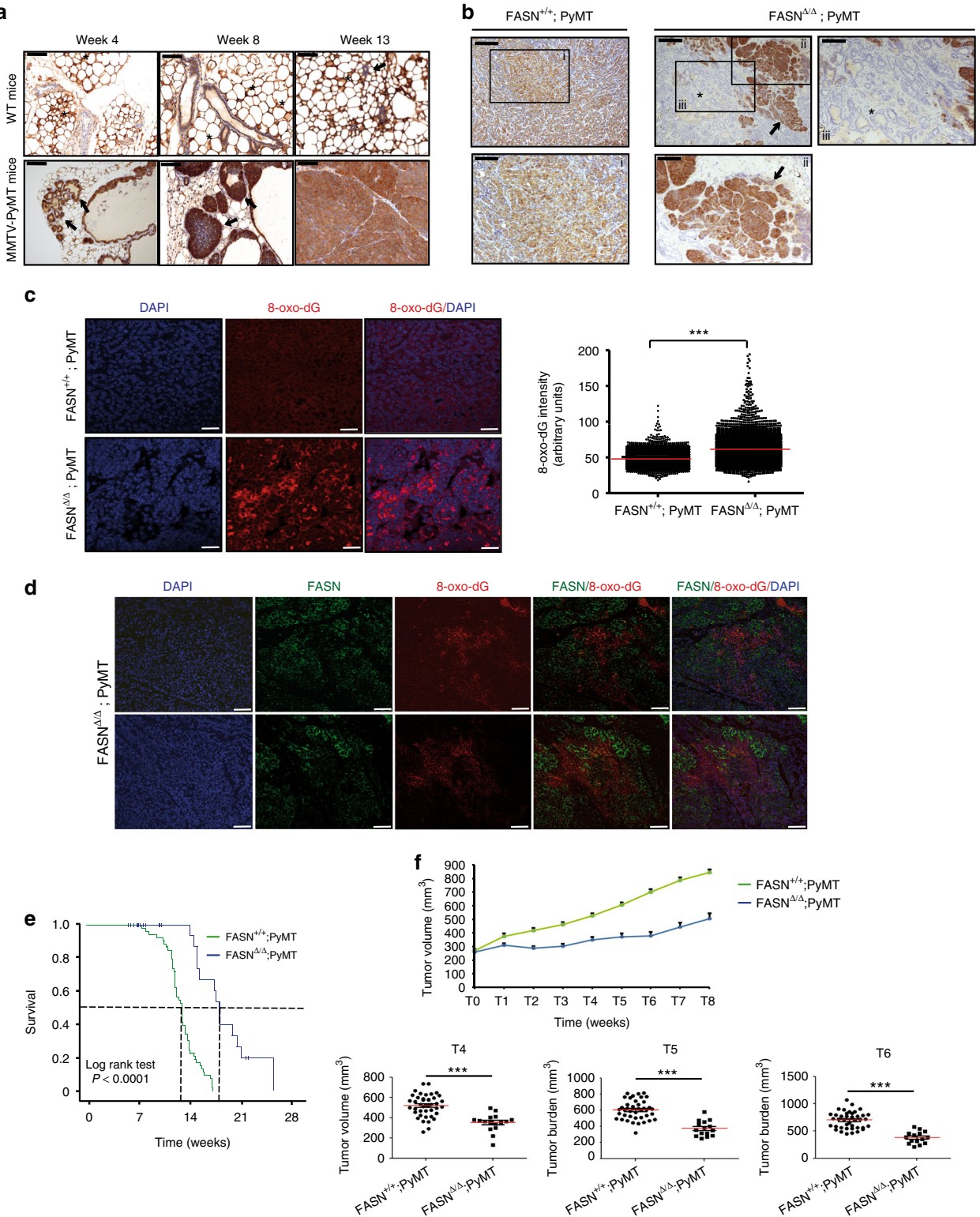

**Cell lines' generation and treatments**. MEFs were isolated from FASN$^{lox/lox}$ embryos (embryos were collected at E12.5). MEFs were grown in Dulbecco's minimum essential media (DMEM, Invitrogen) supplemented with 10% FBS (Sigma) and antibiotics (Sigma) in a humidified incubator at 37 °C with 5% $CO_2$. For immortalization, cultures were retrovirally transduced with SV40 T antigen (in vectors pBABE-hygromycin kindly provided by Manuel Serrano, Spanish National Cancer Research Centre, Madrid). For genetic deletion of FASN, FASN$^{lox/lox}$ MEFs were infected with adeno-Cre (Gene Transfer Vector Core, University of Iowa). Empty adenoviruses (Gene Transfer Vector Core, University of Iowa) were used as a control. MEFs were incubated for 24 h at 37 °C with a reduced volume of culture medium containing the virus at the appropriate concentration, and then refed with fresh medium. Three days post infection, MEFs were used for oncogenic transformation experiments.

HC11_FASN-knockout cells were generated by using the CRISPR/Cas9 All-in One Lentivector System (Applied Biological Materials). Briefly, recombinant lentiviruses were generated in HEK293T cells by co-transfection of 5 μg of single-guide RNA (sgRNA)-encoding plasmids targeting mouse FASN (Cat#K3717105) or scrambled sgRNA (Cat#K010) with 2.5 μg of pVSV-G and 2.5 μg of psPAX2 packaging plasmids (Addgene) with 750 μL of Opti-MEM that contained 30 μL of Lipofectamine 2000 (Sigma). After 48 h, HC11 mammary epithelium cells (ATCC, CRL-3062) were infected, and stable integrants were selected with puromycin (2 μg/mL). HC11 cells were grown in RPMI-1640 medium (Invitrogen)

**Fig. 6** fects of FASN deletion in tumor development. **a** FASN-stained slides from wild-type and MMTV-PyMT FVB mice mammary pads at weeks 4, 8, and 13. FASN is expressed in the cytoplasm of adipocytes (asterisks) in wild-type animals. A weaker FASN expression was observed in breast epithelial ducts (arrows). Conversely, FASN upregulation was observed in the proliferating ducts (week 4), invasive carcinomas arising from the ducts (week 8), and fully developed tumors (week 13) in the MMTV-PyMT animals. Scale bars, 100 μM. **b** Representative images of sections of the mammary pad of FASN$^{+/+}$; PyMT and FASN$^{\Delta/\Delta}$; PyMT animals stained for FASN. The areas negative for FASN have normal or at most proliferative (but not invasive) ducts (asterisks). Invasive growth was only observed in the FASN-positive areas of the FASN$^{\Delta/\Delta}$; PyMT mice (arrows). Scale bars, 200 μm (low magnification) and 100 μm (high magnification; i, ii, iii). **c** Representative fluorescent images of DAPI (blue) and 8-oxo-deoxyguanosine (8-oxo-dG) (red) staining of mammary pad sections of FASN$^{+/+}$; PyMT and FASN$^{\Delta/\Delta}$; PyMT animals. Merged images are shown. The quantification of cellular 8-oxo-dG fluorescence intensity in breast tumors is shown on the right (FASN$^{+/+}$; PyMT, $n = 4500$ cells; FASN$^{\Delta/\Delta}$; PyMT, $n = 6500$ cells). Presented data are the mean values ± SD. ***$P < 0.001$; Student's $t$ test. Scale bars, 50 μM. **d** Representative fluorescent images of DAPI (blue), FASN (green), and 8-oxo-dG (red) co-staining of mammary pad section of FASN$^{\Delta/\Delta}$; PyMT animals. Scale bars, 50 μM. **e** Time to killing due to loco-regional tumor growth of FASN$^{+/+}$; PyMT ($n = 43$) and FASN$^{\Delta/\Delta}$; PyMT ($n = 44$) animals; survival is increased from 90 to 123 days (log-rank $P < 0.0001$). **f** Tumor volume evolution in FASN$^{+/+}$; PyMT ($n = 43$) and FASN$^{\Delta/\Delta}$; PyMT ($n = 44$) animals. Representative tumor burden charts at T4, T5, and T6 weeks (since treatment initiation) are shown below the survival curve (T4: FASN$^{+/+}$; PyMT $n = 42$, FASN$^{\Delta/\Delta}$; PyMT $n = 16$; T5: FASN$^{+/+}$; PyMT $n = 41$, FASN$^{\Delta/\Delta}$; PyMT $n = 15$; T6: FASN$^{+/+}$; PyMT $n = 40$, FASN$^{\Delta/\Delta}$; PyMT $n = 15$). Presented data are the mean values ± SEM. ***$P < 0.001$; Student's $t$ test

supplemented with 10% FBS (Sigma) and antibiotics (Sigma) in a humidified incubator at 37 °C with 5% $CO_2$. For oncogenic transformation experiments, MEFs and HC11 cells were retrovirally transduced with empty (EV), PyMT (kindly provided by Erwin Wagner, Spanish National Cancer Research Centre, Madrid), KRAS (G12D), or HER2 (A775_G776insYVMA)-containing pBABE-puromycin or pBABE-hygromycin vectors as required. Retroviral infections were done by using standard procedures, and infected MEFs and HC11 cells were selected with puromycin (2 μg/mL) or hygromycin (150 μg/mL) as required for at least 5 days.

Dishes with ultralow attachment surface (Corning) were used for suspension and spheroid culture. Identical culture medium was used for monolayer and spheroid culture. For spheroids, $3 \times 10^5$ cells were plated in a 10-cm ultralow attachment dish. For rescue experiments, NAC (Sigma) 1 mM, GSH-MEE, a membrane-permeable derivative of GSH (Santa Cruz) 1 mM, dNTP 100 μM (four deoxynucleotides: dATP, dCTP, dGTP, and dTTP, each at a concentration of 100 mM) (Sigma), soraphen A 200 nM (soraphen A was kindly provided by Dr. Mark Brönstrup, Helmholtz Centre for Infection Research, Braunschweig, Germany), or palmitate 0.5 mM (Sigma) were added to the media 24 h after adeno-Cre infection in monolayer cultures. Sodium citrate (Sigma) or SB204990 (Tocris) was added to the media before PyMT, KRAS, and HER2 overexpression in monolayer cultures. For DMNQ (Sigma) treatment, 10 μM was directly added to FASN$^{lox/lox}$-PyMT MEFs in 3D cultures. To assess the role of exogenous fatty acid in colony formation essentiality first, full medium with FBS was supplemented with the fatty acid uptake inhibitor SSO (25 μM). Second, media was supplemented with 10% fatty acid-free bovine serum albumin (BSA) (Sigma) instead of FBS. Third, cells were treated with FFA-free BSA–palmitate (0.2 mM) conjugate (1:6 molar ratio) in cell culture medium. Spheroid size was determined by measuring the maximum cross-sectional area of individual spheroids by using ImageJ software.

**Animal experiments.** All animal experiments were approved by the Instituto de Salud Carlos III Ethics Committee (PROEX 387/15) and performed in accordance with the guidelines stated in the International Guiding Principles for Biomedical Research Involving Animals developed by the Council for International Organizations of Medical Sciences. For spontaneous tumorigenesis studies, FASN$^{+/+}$; Cre$^+$; PyMT$^+$ and FASN$^{lox/lox}$;Cre$^+$; PyMT$^+$ females were fed tamoxifen diet (Envigo) at 5 weeks of age for early FASN deletion. For the animal treatment, G28UCM was dissolved in 10% dimethyl sulfoxide/90% phosphate-buffered saline (PBS) solution and administered by intraperitoneal injection at the dose of 40 mg/kg per day starting at 5 weeks of age. Body weight was registered weekly. Tumor formation and growth were monitored weekly by using calipers.

Four- to 6-week-old female athymic nude mice (Hsd: athymic nude-Foxn1nu) were purchased from Charles River Laboratories. For mammary fat pad injections, $2 \times 10^6$ cells were resuspended in 50% Matrigel (Corning) and injected (50-μl volume) into female mammary fat pads. Tumor formation and growth were monitored weekly by using calipers. Tumor volumes were calculated by using the formula $V = (D \times d^2)/2$ mm$^3$, where $D$ is the largest diameter and $d$ is the shortest diameter. Mice were euthanized in a $CO_2$ chamber when reaching humane end point (1500 mm$^3$). Tumors were dissected from the front limb mammary fat pad and fixed (10% formalin solution) for histological examination (formalin-fixed paraffin-embedded). % Tumor growth was calculated by using the following formula: % Tumor growth = $[1 - (T_F/T_0)A/(T_F/T_0)V] \times 100$, where $T_F$ is the time point analyzed, $T_0$ is the initial time, A is the tumor measurement corresponding to FASN$^{\Delta/\Delta}$; PyMT mice, and V is the tumor measurement from FASN$^{+/+}$; PyMT mice.

**Cell viability assay.** For cell viability assays, FASN$^{+/+}$-PyMT MEFs and HC11 cells were seeded ($5 \times 10^3$ cells/well) in 96-well plates for 24 h and subsequently replaced with fresh medium with vehicle (control) or increasing concentration of sodium citrate (Sigma) (1, 2.5, 5.0, and 10 mM) or SB204990 (Tocris)

Bioscience) (10, 20, 30, 40, 60, and 100 μM). Viability assays were performed in triplicates. After 24, 48, and 72 h, cell viability was analyzed by using CellTiter-Glo Luminescent Cell Viability Assay (Promega) following the manufacturer's instructions.

**Soft-agar assay.** Anchorage-independent growth was tested as follows: a base layer of 0.9% agar in complete medium (DMEM plus 10% FBS) was plated in six-well plates and allowed to solidify. Next, wells were overlaid with $15 \times 10^4$ cells per well in a 0.4% agar. The plates were incubated at 37 °C, 5% $CO_2$ for 3–4 weeks and checked every 2–3 days for colony formation. Colonies were counted in the entire dish by using a microscope.

**Immunohistochemistry.** For routine histological analysis, tissues were fixed in 10% buffered formalin (Sigma) and embedded in paraffin. Immunohistochemical staining was performed on 3- to 4-μm paraffin sections. Immunohistochemistry was performed by using an automated protocol developed for the DISCOVERY XT-automated slide-staining system (Ventana Medical Systems Inc.). All steps were performed on this staining platform by using validated reagents, including deparaffinization, antigen retrieval (cell conditioning), and Ab incubation and detection. Anti-FASN primary Ab (1:50, clone C20G5; 3180, Cell Signaling), anti-collagen type IV Ab (1:250, CL50451AP; Cederlane), anti-Ki67 Ab (clone MIB-1, M7240, Dako), anti-pan-RAS Ab (1:750, clone Ab-3; OP40, Merck), and anti-HER2 (c-erB2) Ab (1:400, clone SP495; E12424, Spring Bioscience) were optimized to identify optimal conditions for cell conditioning, dilution, incubation, and detection. Localization of the primary Abs was accomplished by using an appropriate biotinylated secondary Ab, followed by incubation with streptavidin–horseradish peroxidase and diaminobenzidine system. Slides were digitalized and analyzed by using the ZEISS Zen 2.3 Imaging Software (Zeiss).

**Western blotting.** Cells were washed 2× with PBS and harvested in cold radio-immunoprecipitation buffer (Sigma) containing 1% protease and phosphatase inhibitor cocktail (Halt EDTA-free; Thermo Scientific). Cell lysates were incubated at 4 °C for 15 min, sonicated for 15 min, and clarified by centrifugation at 14,000 × $g$ at 4 °C for 30 min. Protein concentration was estimated by a colorimetric assay (660-nm protein assay; Pierce) following the manufacturer's instruction. Twenty micrograms of proteins per sample were loaded on 10% sodium dodecyl sulfate-polyacrylamide gel electrophoresis gel and transferred to nitrocellulose membranes for further processing. Five percent of BSA was used to block the membrane for 60 min at room temperature, followed by overnight incubation at 4 °C with the primary Abs. The following primary Abs were used: FASN (1:1000, clone: C20G5; 3180, Cell Signaling), PyMT (1:2500, NB100-2749, Novus Biological), K-Ras (1:1000, clone: F234; sc-30; Santa Cruz), HER2 (1:1000, 2242, Cell Signaling), IDH1 (1:1000, clone: D2H1, 8137, Cell Signaling), β-actin (1:5000, clone AC-15; A5441, Sigma), γ-tubulin (1:2500, clone GTU-88; T6557, Sigma), and vinculin (1:2000, clone: hVIN-1; V9131, Sigma). Membranes were incubated with appropriate peroxidase-conjugated secondary Abs (Sigma). Bands were visualized by the enhanced chemiluminescence method (Lumi-LightPlus Detection Kit; Roche). The ImageJ software was used to compare the density of bands. The uncropped and unprocessed scans of the blots were provided in Supplementary Fig. 11.

**Immunofluorescence.** Immunohistofluorescence was performed on formalin-fixed mouse tissues. Five-micrometer paraffin-embedded sections were deparaffinized and rehydrated in graded alcohol series. After rehydration, the paraffin sections were boiled in a microwave oven for epitope retrieval in sodium citrate buffer (10 mM [pH 6]) for 15 min. Sections were equilibrated in water and incubated with blocking solution (5% normal goat serum in TBS-T [Tris-buffered saline with Tween-20]) for 1 h at room temperature. Later it is washed with 1× PBS and

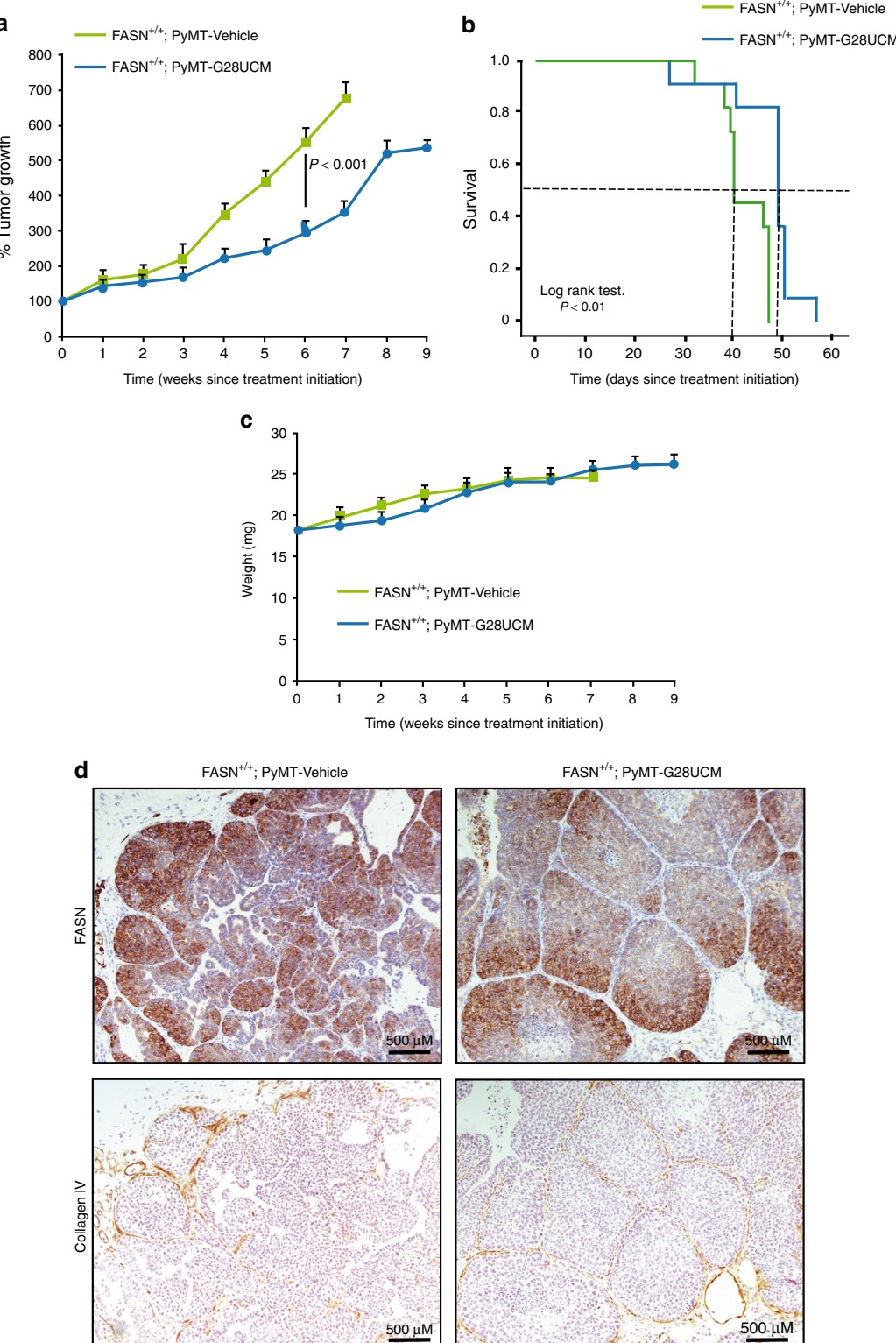

**Fig. 7** FASN inhibition prevents the development of invasive ductal carcinoma. **a** Inhibitory effects of G28UCM on tumor growth in MMTV-PyMT animals. Presented data are the mean values ± SEM ($n = 11$ mice per group). ***$P < 0.001$; Student's $t$ test. **b** Kaplan–Meier survival curves of mice treated with either vehicle ($n = 11$ mice) or G28UCM agent ($n = 11$ mice). Survival was calculated from the time of treatment initiation. Survival is increased in G28UCM-treated mice from 40 to 49 days (log-rank $P < 0.01$). **c** Average animal weight during vehicle ($n = 11$ mice) or treatment ($n = 11$ mice). **d** Upper panels: FASN staining in vehicle- or G28UCM-treated breast tumors. Lower panels: collagen IV staining (basement marker) reveals that G28UCM-treated MMTV-PyMT animals develop only noninvasive ductal in situ carcinoma proliferations. Representative images were obtained for each treatment. Scale bar, 500 μm

incubated with anti-8-oxo-dG Ab (1:250, clone: 2E2; 4354-MC-050, Trevigen) at a concentration of 1:250 at 4 °C o/n in a humidified chamber. Sections were washed twice in TBS-T for 10 min and incubated with Alexa Fluor 594 donkey anti-mouse immunoglobulin G (IgG) (1:200; Thermo Fisher Scientific) secondary Ab for 1 h at room temperature. Sections were counterstained with DAPI (4′,6-diamidino-2-phenylindole) and mounted in ProLong Gold antifade reagent (Life Technologies). Digital images were taken by a TCS SP5 confocal microscope (Leica Microsystems) at a resolution of 757.6 nm/px. Images were captured by using a ×20/0.7NA dry objective, and the optical section was 3.665 µm. Images were analyzed with Definiens Developer XD software with a customized script for detection and intensity quantification. For immunofluorescence double staining with anti-FASN and anti-8-oxo-dG, paraffin sections were incubated with rabbit anti-FASN (dilution 1:50, Cell Signaling) and mouse anti-anti-8-oxo-dG (1:250, Trevigen) at 4 °C o/n in a humidified chamber. Sections were washed twice in TBS-T for 10 min and incubated with Alexa Fluor 488 goat anti-mouse IgG (1:200; Thermo Fisher Scientific) and Alexa Fluor 555 goat anti-rabbit IgG (1:200; Thermo Fisher Scientific) secondary Ab for 1 h at room temperature to probe 8-oxo-dG and FASN, respectively. Immunofluorescence images were taken by TCS SP5 confocal microscope (Leica Microsystems) at a resolution of 757.6 nm/px.

**Extracellular flux analysis.** OCR and ECAR were determined by using the XF96 Extracellular Flux Analyzer (Seahorse Bioscience) following the manufacturer's protocol. For monolayer culture, MEFs and HC11 cells were seeded at a density of $1 \times 10^4$ cells/well (optimal cell-seeding density was optimized in preliminary experiments) in an XF96-well microplate 12 h before experiments. For spheroid assays, cells were cultured in ultralow attachment surface plates for at least 3 days. Spheroids were disaggregated, counted, and plated into the XF96-well microplate. The microplate was pre-coated with cell adhesive (Cell-tak Cell and tissue adhesive; Corning) to maintain cells in the proper location. For OCR determination, assays were initiated by replacing the media with XF assay media supplemented with fresh sodium pyruvate (1 mM), glutamine (2 mM), glucose (0.5 mg/L), and 10% FBS (pH 7.4), and were equilibrated in non-CO2 incubator at 37 °C for 1 h. Cells were then placed in the instrument, and oxygen consumption was recorded for almost 90 min. Seahorse Analyzer uses a cartridge with 96 optical fluorescent O2 sensors to measure OCR (pmol/min). OCR was measured simultaneously in all wells three times at each step, and a minimum of eight replicates were utilized per condition in any given experiment. Basal OCR was measured, followed by sequential treatment of oligomycin A (1 µM), carbonylcyanide-$p$-trifluoromethoxyphenylhydrazone (FCCP) (0.9 µM), and rotenone/antimycin (1 µM) (optimal reagent concentrations were determined in preliminary experiments). For ECAR determination, after 24 h of incubation, the media was changed to assay medium (XF base medium supplemented with 2 mM glutamine), and cells were incubated in a non-CO2 incubator at 37 °C for 1 h before the assay. Injections of glucose (10 mM), oligomycin (1 µM), and 2-DG (100 mM) were diluted in the assay medium and loaded into ports A, B, and C, respectively. The machine was calibrated, and the assay was performed by using glycolytic stress test assay protocol as suggested by the manufacturer (Seahorse Bioscience). Each treatment was measured every 8 min (4 min of measurement) three times. All compounds and materials were obtained from Seahorse Bioscience. Protein concentrations in each well were determined with the BCA method (Pierce) in cell lysates. Respiration and acidification rates are presented as mean values ± SEM.

**Mitochondrial membrane potential measurements.** Cells were loaded with 75 nM of the fluorescent potential-dependent indicator dye (Abcam) for 30 min at 37 °C. The TMRE (tetramethylrhodamine, ethyl ester) dye is a cell-permeant, positively charged, red-orange dye that readily accumulates in active mitochondria due to their relative negative charge. The mitochondrial marker Mitotracker green (Thermo Fisher Scientific) (200 nM) stains mitochondria independently of the mitochondrial membrane potential. This dye was used as a positive marker of mitochondria. Uncoupler FCCP (20 µM) was added to FASN$^{lox/lox}$-PyMT cells as a negative control. Digital images were taken by a TCS SP5 confocal microscope (Leica Microsystems) at a resolution of 757.6 nm/px. Images were captured by using a ×20/0.7NA dry objective, and the optical section was 3.665 µm. Images were analyzed by using a custom-developed routine programmed on Definiens Developer v2.5 software (Definiens, Germany); on this routine the identification of mitochondrial regions was done by using the green channel, and the membrane potential was quantified by using the regions from these segmented objects on the red channel. Some samples incubated with 20 µM FCCP, an uncoupler of electron transport and oxidative phosphorylation, for 10 min prior to staining with TMRE, served as a positive control for depolarized mitochondria.

**ATP citrate lyase activity.** MEFs were maintained in ultralow attachment surface plates for 72 h. ATP-citrate lyase activity was measured by using the mouse ATP-Citrate Lyase ELISA Kit (LifeSpan BioSciences) following the manufacturer's instructions. Optimal cell number was optimized in preliminary experiments. Briefly, cultured cells ($4 \times 10^7$) were collected by centrifugation and washed three times with PBS. Cell pellets were then resuspended in PBS and lysed by ultrasonication. Cellular debris was removed by centrifugation. One hundred microliters of supernatant was collected for assaying. After processing, optical density of each well was determined by using a microplate reader set to 450 nm. Four replicates were utilized per condition in any given experiment.

**Mitochondria isolation and solubilization.** Mitochondria were isolated according to a previously published protocol[67]. Briefly, cultured cells ($4 \times 10^7$) were collected by using trypsin/EDTA solution, and frozen at −80 °C to facilitate cell breakage. Cell pellets were homogenized by using Potter–Elvehjem glass-teflon homogenizer backed with electrical support, and adding about ten cell pellet volumes of homogenizing buffer A (83 mM sucrose, 10 mM MOPS (3-($N$-morpholino)propanesulfonic acid)), pH 7.2). An equal volume of buffer B (250 mM sucrose, 30 mM MOPS, pH 7.2) was added, and samples were centrifuged at $1000 \times g$ during 5 min to remove unbroken cells and nuclei. Mitochondria were collected from the supernatant by centrifuging at $12,000 \times g$ during 3 min. The pellets were washed with buffer C (320 mM sucrose, 1 mM EDTA, and 10 mM Tris-HCl, pH 7.4), and centrifugation was repeated at the same conditions. Mitochondrial pellets were suspended in buffer D (1 M 6-amiohexanoic acid, 50 mM Bis-Tris-HCl, pH 7.0), and the membrane proteins were solubilized with digitonin (4 g/g) and incubated for 5 min in ice. A 30-min centrifugation at $13,000 \times g$ was performed, and then the supernatant containing the solubilized mitochondria was collected.

**Blue native gel electrophoresis and transfer.** The extracted and solubilized mitochondria were separated by electrophoresis in native gels (NativePAGE Novex 3–12% Bis-Tris Protein Gels, Thermo Scientific) in order to evaluate the arrangement of the complexes in the electron transport chain. The mitochondrial samples were loaded in 10% (v/v) loading buffer (5% (v/v) Serva-Blue G; 1 M 6-aminohexanoic acid). The electrophoresis was carried out at 100 V for 1 h in the presence of blue cathode buffer (50 mM tricine; 15 mM Bis-Tris/HCl; 0.02% (w/v) Coomassie Brilliant Blue G, pH 7) in the cathode chamber, and anode buffer (50 mM Bis-Tris, pH 7) in the anode chamber, followed by buffer replacement (removal of Coomassie blue dye) in the cathode chamber and electrophoresis at 40 V overnight at 4 °C. The gel containing the separated proteins was then incubated at 4 °C in carbonated transfer buffer (10 mM NaHCO3; 3 mM Na2CO3·10 H2O, pH 9.5–10) for 20 min with agitation to ensure that the gel was fully saturated with said transfer buffer. The transfer was then carried out onto polyvinylidene difluoride membranes (Thermo Scientific) (previously activated with methanol) at 60 V for 90 min at 4 °C in the Mini-Transblot system (Bio-Rad, USA). The membranes were then subjected to normal western blot protocol by using anti-NDUFS1 primary antibody (1:500) (Sc-50132, Santa Cruz) and anti-goat (1:10000) (ab6741, Abcam) as a secondary antibody.

**In-gel activity of complex I.** The activity of complex I was evaluated by using in-gel activity assay. For this, after electrophoresis, gels were washed in running buffer to eliminate any rest of loading buffer. Then, gels were incubated in fresh activity solution (0.1 M Tris-HCl, pH 7.4, 1 mg/mL nitro blue tetrazolium chloride, and 0.14 mM NADH) at room temperature during 30–60 min[68]. Reaction was stopped by washing the gels with distilled water. Apparition of blue–purple precipitate indicates the activity of NADH oxidase, one of the activities of complex I, confirming by this manner the localization of this complex.

**Pyruvate dehydrogenase activity.** The PDH activity was determined in isolated mitochondria from fresh cells (Mitochondria Isolation Kit, BioVision) by using a coupled enzyme reaction, which results in a colorimetric product proportional to the enzymatic activity (BioVision). Acetyl-CoA (Sigma) was added directly to the mitochondrial lysates (10 mM) before the assay.

**Cell growth assays.** Cell proliferation was measured by counting cells after trypsinization in 2D cultures. In 3D cultures, spheroids were desegregated before counting. The average spheroid size was determined by measuring the maximum cross-sectional area of individual spheroids by using the ImageJ software. A minimum of ten images per condition were used.

**Plasma triglyceride determination.** Plasma levels of triglycerides were measured by using an ABX Pentra Clinical Chemistry benchtop analyzer (Horiba ABX, Montpellier, France) using the ABX Pentra Triglycerides CP reagent following the manufacturer's instructions (Horiba ABX, Montpellier, France). Briefly, blood was collected by using heparin as the anticoagulant followed by 10 min of centrifugation at $1500 \times g$ at room temperature. The yellow plasma layer was transferred to another tube. Three microliters of plasma was used per test, in triplicates.

**Oil Red-O staining.** Tissues were washed with PBS and then fixed in 10% buffered formalin for 1 h at room temperature. After washes in PBS, cells were stained for 30 min at room temperature with a filtered Oil Red-O (Sigma) solution (0.5% Oil Red-O in isopropyl alcohol), washed again in PBS, and visualized under an inverted microscope (Olympus).

**RNA extraction and quantitative RT-PCR.** Total RNA was extracted from cells with TRIzol reagent (Thermo Fisher Scientific) in accordance with the

manufacturer's instructions. Briefly, the same quantity of total RNA was retro-transcribed to complementary DNA (cDNA) by using the Quantitect Reverse Transcription Kit (Qiagen, Valencia, CA, USA) (2 min at 42 °C, 15 min at 42 °C, and 3 min at 95 °C). One microliter of the resulting cDNA was placed in a 384-well plate with 5 µl of Syber Green (Applied Biosystems-Life Technologies, Grand Island, NY, USA) and 2 µl of the corresponding primers in a final volume of 10 µL. PCR amplification was performed by using the Applied Biosystems Prism 7900HT Sequence Detection System (Life Technologies; Grand Island, NY, USA) under the following thermal cycler conditions: 2 min at 50 °C, 10 min at 95 °C, and 40 cycles (15 s at 95 °C and 1 min at 60 °C). To quantify transcription, the mRNA expression levels of the target genes were normalized to β-actin (GGCTCCTAGCACCAT-GAAGA—forward; CCACCGATCCACACAGAGTA—reverse). The primers used are as follows: (sequence (5′–3′): FASN-CTGCCACAACTCTGAGGACA (forward) and CGGATCACCTTCTTGAGAGC (reverse); Glut1-GCCCCCAGA AGGTTATTGA (forward) and CGTGGTGAGTGTGGTGGAT (reverse); Glut4-ACATACCTGACAGGGCAAGG (forward) and CGCCCTTAGTTGGTCAGAAG (reverse)).

All samples were run in triplicates, and relative quantification (RQ) was calculated following the ΔCt method: $RQ = 2^{-\Delta Ct}$, where ΔCt is the difference between the Ct of the gene of interest and the Ct of the endogenous gene control β-actin. In addition, in knockdown experiments, RQ was normalized as $RQ = 2^{-\Delta\Delta Ct}$, where ΔΔCt is the difference between the ΔCt in knockdown cells and the ΔCt in control cells.

**NAD$^+$ and NADH measurements.** Total intracellular NAD$^+$ and NADH were measured by using the EnzyChrom NAD$^+$/NADH Assay Kit (BioAssay Systems) following the manufacturer's instructions.

**NADPH assay.** Total cell lysates or mitochondrial lysates were used to determine NADPH levels in 3D culture conditions. Briefly, $20 \times 10^6$ cells were cultured in ultralow attachment surface plates for at least 72 h before NADPH quantification. Mitochondrial Isolation Kit (BioVision) was used to obtain the mitochondrial lysates following the manufacturer's instructions. NADPH levels were assessed in total cell lysates and mitochondrial lysates by using the NADP/NADPH Assay Kit (abcam) following the manufacturer's instructions.

**ATP and ADP content.** ATP and ADP levels were measured by using the Enzy-Light ADP/ATP Assay Kit (BioAssay Systems) following the manufacturer's instructions.

**Apoptosis and ROS measurements.** For apoptosis assay, MEFs were stained with DAPI and Annexin V (cat no. 556547, FITC Annexin V Apoptosis Detection Kit I, BD Pharmingen™) at room temperature for 5 min and analyzed by flow cytometry by using a FACSCalibur flow cytometer (FACScalibur, Becton Dickinson). A representative example of gating strategy used to characterize FASN$^{lox/lox}$-PyMT MEFs in apoptosis in 2D cultures, by using co-staining with Annexin-V and DAPI is shown in Supplementary Fig. 1h. Mitochondria-mediated ROS generation was detected with the mitochondrial superoxide indicator MitoSOX-Red (Life Technologies). For 2D measurement of mitochondrial ROS, MEFs, and HC-11, cells were trypsinized and resuspended in DMEM and RPMI medium with 10% FBS. After centrifugation, cells were washed twice in PBS and stained with 3 µM of Coomassie-Red (Life Technologies) for 20 min in PBS for 30 min at 37 °C. Subsequently, the cells were washed twice in PBS, followed by analysis on a FACS Calibur flow cytometer (Becton Dickinson). For 3D measurement, MEFs and HC11 cells were cultured in ultralow attachment surface plates for at least 72 h before ROS quantification. After incubation with MitoSOX-Red, cells were then washed in PBS with 5 mM EDTA to prevent aggregation of cells for FACS analysis. DAPI staining was used for cell death detection. Negative controls with no stain were used in all experiments. The data obtained from flow cytometer were analyzed by using the FlowJo software (Tree Star, Ashland, OR). Gating strategy to evaluate ROS content by using MitoSOX-Red is shown in Supplementary Fig. 3e.

**Isotope-tracing experiments.** For isotope-labeling experiments, glucose or glutamine were replaced with their U-$^{13}$C-labeled forms (Cambridge Isotope Labs), U-$^{13}$C$_6$-glucose, and U-$^{13}$C$_5$-glutamine, respectively. For all $^{13}$C-tracing experiments, cells were maintained in the labeled medium for 8 h. For 3D culture conditions, cells were seeded in ultralow attachment surface dishes (Corning).

**Metabolite extraction.** Metabolites were extracted into the extraction solvent by adding 220 µL of cold methanol in 0.1% formic acid. Samples were vortexed for 30 s and immersed in liquid N2 to disrupt cell membranes, followed by 30 s of bath sonication. These two steps were repeated three times. Four hundred and forty microliters of dichloromethane was added to the resulting suspension, followed by 140 µL of cold water. Cell lysates were vortexed before centrifugation (5000 × g, 15 min at 4 °C), and the polar (aqueous) and nonpolar (lipidic) phase were separated and carefully transferred into new tubes. The polar sample was frozen, lyophilized, and stored at 80 °C until further mass spectrometry (MS) analysis. The

nonpolar phase was dried under a stream of nitrogen gas before nuclear magnetic resonance (NMR) analysis.

**NMR analysis.** Polar (aqueous) samples were analyzed by using positional enrichment by proton analysis (PEPA). PEPA detects the position of carbon label in isotopically enriched metabolites and quantifies fractional enrichment by indirect determination of $^{13}$C-satellite peaks by using 1D-H-NMR spectra.

$^{13}$C NMR spectra of nonpolar (lipidic) samples were recorded at 300 K on an Avance III 600 spectrometer (Bruker, Germany) operating at a carbon frequency of 150.93 MHz by using a 5-mm CPTCI triple-resonace ($^1$H, $^{13}$C, and $^{31}$P) gradient cryoprobe. Power gate-decoupled $^{13}$C pulse experiments ("zgpg" in Bruker® pulse sequence) were carried out by using WALTZ-16 scheme with proton presaturation and relaxation delay of 10 s. The 90° pulse length was 14.75 ms. The spectral width was 36 kHz (240 p.p.m.), and a total of 256 transients were collected into 64k data points for each $^{13}$C spectrum. The exponential line broadening applied before Fourier transformation was 1 Hz. The frequency domain spectra were phased and baseline-corrected and referenced to the chemical shift of TSP signal at 0 p.p.m. by using the TopSpin software (version 2.1, Bruker). After preprocessing, specific $^{13}$C NMR peaks around 14.17 p.p.m. characteristic of the terminal CH$_3$ fatty acid were integrated by using the AMIX 3.9 software package.

**Gas chromatography–MS analysis.** We incubated the lyophilized polar extracts with 50 µL of methoxyamine in pyridine (40 µg/µL) for 45 min at 60 °C. To increase volatility of the compounds, we silylated the samples by using 25 µL of N-methyl-N-trimethylsilyltrifluoroacetamide with 1% trimethylchlorosilane (Thermo Fisher Scientific) for 30 min at 60 °C.

A 7890A GC system coupled to a 7200 QTOF or 7000 QqQ MS (Agilent Technologies, Palo Alto, CA) was used for isotopolog determination. Derivatized samples were injected (1 µL) in the gas chromatograph system with a split inlet equipped with a J&W Scientific DB5-MS + DG stationary-phase column (30 mm × 0.25 mm i.d., 0.1-µm film, Agilent Technologies). Helium was used as a carrier gas at a flow rate of 1 mL/min in constant flow mode. The injector split ratio was adjusted to 1:5, and oven temperature was programmed at 70 °C for 1 min and increased at 10 °C/min to 325 °C. The ionization performed was positive chemical ionization with isobutene as reagent gas. Mass spectral data on the 7200 QTOF were acquired in full-scan mode from m/z 35 to 700 with an acquisition rate of 5 spectra per second. Mass spectral data on the 7000 QqQ were acquired in scan mode monitoring selected ion clusters of the different metabolites.

**Liquid chromatography–MS analysis.** Polar samples were analyzed by using a UHPLC system (1290 series, Agilent Technologies) coupled to a 6550 ESI-QTOF MS (Agilent Technologies) operated in positive (ESI+) and negative (ESI−) electrospray ionization mode. Metabolites were separated by using C18-RP (ACQUITY UPLC HSS T3 1.8 µm for ESI+ and BEH 1.7 µm for ESI−, Waters) chromatography at a flow rate of 0.4 mL/min. The solvent system in C18-RP ESI+ was A = 0.1% formic acid in water, and B = 0.1% formic acid in acetonitrile. The solvent system C18-RP ESI− was A = 1 mM ammonium fluoride in water (pH ~7) and B = acetonitrile, as previously reported[69]. The linear gradient elution started at 100% A (time 0–2 min) and finished at 100% B (10–15 min). The injection volume was 5 µL by using a flow rate of 0.4 mL/min. ESI conditions: gas temperature, 150 °C; drying gas, 13 L/min; nebulizer, 35 psig; fragmentor, 175 V;skimmer, 65 V. The instrument was set to acquire over the m/z range 100–1500 in full-scan mode with an acquisition rate of 4 spectra/s. Tandem MS (MS/MS) was performed in targeted MS/MS mode, which requires that the masses of selected metabolites be specified on an inclusion list to ensure that they are subjected to a second stage of mass selection in an MS/MS analysis. MS/MS experiments were acquired in order to confirm the identity of unlabeled and labeled peaks detected by the software geoRge[70]. We used a default isolation width (the width half-maximum of the quadrupole mass bandpass used during MS/MS precursor isolation) of 4m/z, and the collision energy was fixed at 20 eV.

**[$^{14}$C]acetate incorporation into lipid fractions.** MEFs were grown in 12-well plates to 70–80% confluence. MEFs were treated with 1 mCi/mL sodium [1,2-$^{14}$C] acetate (PerkinElmer) for 6 h. After two washes with ice-cold PBS, cells were lysed with 0.6 mL of MeOH solution (MeOH:H$_2$O = 2.5:1). CHCl$_3$ (0.4 mL) was added to the lysate and mixed by vortexing for 30 s. Lysates were then centrifuged for 5 min at 1000 r.p.m. for phase separation. The lipid-soluble fraction was collected as the lower layer. Fractions were counted for radioactivity.

**Palmitate oxidation.** For assessment of palmitate metabolism, FASN$^{lox/lox}$-PyMT and FASN$^{\Delta/\Delta}$-PyMT MEFs were cultured in monolayer (2D) or spheroid (3D) culture conditions for 48 h. After that MEFs were washed with PBS 1× twice and they were incubated in the media containing 20 mM HEPES, 140 mM NaCl, 16.1 mM KCl, 5.1 mM MgSO$_4$, 2.7 mM CaCl$_2$, 1.2 mM L-carnitine, 2% FFA-free BSA, 0.8 mM palmitate, and 2 µCi [$^{14}$C]palmitate (PerkinElmer—NEC534050UC) with pH adjusted to 7.4. For rescue experiments, soraphen A was added to the cells (200 mM). Following a 4-h incubation period, 1 mL of the medium was transferred to a 1.5-mL tube, the cap of which housed a Whatman (GF/B) filter paper disc that

had been presoaked with 200 µL of 1 M KOH. [$^{14}$C]O$_2$ trapped in the media was then released by acidification of media by using 60% (vol/vol) perchloric acid and gentle agitation of the tubes at 37 °C for 2 h. Afterward, the filter paper disc was removed from the cap and placed into 2 mL of scintillation liquid. The remaining cells were subsequently washed with 2 mL of PBS and lysed in 1 mL of 1 M NaOH before being transferred into 4 mL of scintillation liquid. Samples were then subjected to liquid scintillation counting (Wallac 1409 Liquid Scintillation counter, PerkinElmer, Waltham, MA, USA). The protein content of samples was measured in triplicates by using the Bradford assay.

**siRNA knockdown of IDH1**. FASN$^{+/+}$-PyMT MEFs were plated into a 10-cm dish. After 24 h, the cells were transfected with 50 nM of either siRNA specific for mouse IDH1 (cat. no. L-045049-00-0005, Dharmacon) or scrambled siRNA (cat. no. D-001810-01-05, Dharmacon) by using DharmaFECT transfection reagents (Dharmacon). After 48 or 72 h, the cells were processed for spheroid formation capacity and OCR assay, and protein was harvested for analysis by immunoblot (see the "Western blotting" section).

**Statistical analysis**. Experimental data are presented as the mean ± standard deviation or mean ± standard error mean. Data were analyzed by Student's $t$ test to compare means of two groups by using the GraphPad Prism® Software (La Jolla, CA). *$P < 0.05$, **$P < 0.01$, and ***$P < 0.001$. Determinations of the significance of the differences in survival outcomes between FASN$^{+/+}$; PyMT and FASN$^{\Delta/\Delta}$; PyMT mice were calculated by Kaplan–Meier survival curve comparisons, as well as the $P$ values derived from log-rank test. Analysis was performed by using the SPSS software (version 19).

**Reporting summary**. Further information on research design is available in the Nature Research Reporting Summary linked to this article.

## Data availability

The data that support the findings of this study are included in this paper and available from the corresponding author upon request.

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

## Acknowledgements

M.Q.F. is a recipient of the following grants: FIS PI13/00430 and FIS PI16/00354 funded by the Instituto de Salud Carlos III (ISCIII) and co-funded by the European Regional Development Fund (ERDF) and AECC Scientific Foundation (Beca de Retorno 2010). R.C. is a recipient of the following grants: FIS PI11/00832 and FIS PI14/00726 funded by the Instituto de Salud Carlos III (ISCIII) and co-funded by the European Regional Development Fund (ERDF), II14/00009 and PIE15/00068 from the Ministerio de Sanidad, Spain. N.S.C. is a recipient of an NIH grant (5R35CA197532). O.Y.T. is a recipient of the grants BFU2014-57466 from the Ministerio de Economía y Competitividad (MINECO). J.P.B. is funded by MINECO (SAF2016-78114-R), Instituto de Salud Carlos III (RD12/0043/0021), Junta de Castilla y León (Escalera de Excelencia CLU-2017-03), Ayudas Equipos Investigación Biomedicina 2017 Fundación BBVA, and Fundación Ramón Areces. This study was partially supported by the generous donations from Fundación CRIS Contra el Cáncer and AVON Spain. We thank Drs. Erwin Wagner and Nabil Djouder for their critical review of the paper.

## Author contributions

M.Q.F. and M.J.B. designed most of the experiments. M.J.B. and V.J.R. performed most of the experimental work. S.S., J.C., and O.Y. acquired and analyzed the data from the MS and magnetic resonance experiments. R.C. and N.S.C. collaborated with M.Q.-F. and M.J.B. in the experimental design and data interpretation. M.L.L.-R., J.G.-C., and R.C. developed and synthesized UCMG28. J.P.B. and I.L.-F. designed and executed, respectively, the experiments related with mitochondrial electron transport chain organization. All authors contributed to writing the paper, figure descriptions, and approval of the final version.

## Competing of interests

The authors declare no competing interests.
