## [Peer Review File · Nature Communications]

Reviewers' comments:

Reviewer #1: Cancer metabolism (Remarks to the Author):

In this study, Bueno et al. show that FASN is required for cell transformation, is essential to maintain low ROS levels and thereby promotes anchorage independent growth. Interestingly, authors show that the role of FASN as a promoter of oncogenic transformation is not dependent on its synthetic products, but mainly relies on an impairment of IDH-1-dependent reductive carboxylation, previously described to be essential for 3D growth (Jiang et al., Nature, 2016). Indeed, authors conclude that FASN deletion induces acetyl-CoA accumulation which consequently inhibits ATP citrate lyase and thereby promotes citrate/isocitrate accumulation. This later accumulation limits IDH1-dependent reductive carboxylation observed in 3D growth and which is needed to supply mitochondria with reduced equivalent to quench excess of ROS produced in mitochondria following spheroids formation. Authors show robust in vitro metabolic tracing data and perform several in vivo models to validate the role of FASN as a promoter of tumor growth.

Major Comments :

Figure S1: Levels of FASN are increased in MEFs following PyMT infection. This increase is not observed following KRAS or HER. Could this increase of FASN participate to the ability of MEFs to transform upon PyMT? In other words do levels of FASN impact on transformation by PyMT? Levels of RAS are higher in FASN^{lox/lox} MEFs compared to FASN^{Δ/Δ} MEFs. As FASN^{lox/lox} MEFs proliferate more (Fig1C) they probably are easier to infect and integrate more retroviral particles which could explain the increased RAS level. So one could conclude that increased RAS levels in FASN^{lox/lox} MEFs participate to the increased clonogenic ability of these cells (Fig S1C) and could be responsible to the rapid tumor growth at T3-T4 observed in FigS6B compared to the slower tumor growth at T8 for FASN^{Δ/Δ}KRAS MEFs. Quantification of western blots (Fig S1) would help.

Figure S1E: HC11FASNKO cell lines are generated by the Crispr-C9 technology. sgRNA targeting human FASN were used. It appears that HC11FASN^{wt} cells were not infected by recombinant lentivirus containing control sg. Such control is missing. Authors should compare both lentivirus infected HC11 cells with or without FASN.

Figure S2A: Authors mention that the decrease in PDH activity in FASN^{Δ/Δ}PyMT MEFs compared to FASN^{lox/lox}PyMT MEFs is phenocopied by adding acetylCoA to FASN^{lox/lox}PyMT MEFs which is not clearly illustrated in the panel S2A where no statistic are illustrated.

Figure 2B. NADPH production is increased in FASN^{Δ/Δ}PyMT MEFs compared with FASN^{lox/lox}PyMT MEFs in 2D condition. Is this increase due to low ROS levels in FASN^{Δ/Δ}PyMT MEFs or vice-versa? Authors mention that, in 3D, there would be insufficient intra-mitochondrial reduced equivalents which are usually consumed by excess of unquenched ROS produced during the 2D to 3D transition. NADPH measurements in 3D should be performed and added in Figure 4 or 5. Moreover, authors should better explain why accumulation of unquenched ROS during 2D to 3D transition would be followed by impaired mitochondrial respiration, as authors write that lack of ROS production in 2D in FASN^{Δ/Δ}PyMT MEFs explains the low OCR observed (lines 165 to 168). These 2 different conclusions in 2D and 3D conditions are confusing. Collaboration with expert in ROS and OXPHOS mechanisms such as Dr N. Chandel, co-authors of this study, should help to make these points cleared up.

Figure 3E : Authors should illustrate other TCA intermediates such as fumarate, malate.

Figure 4 : Authors should examine in MEFs if chemical FASN inhibitor recapitulates some of the features of the FASN^{Δ/Δ}PyMT phenotype in several 3D assays in order to strengthen their in vivo data using the G28UCM compound.

Figure 6 : Authors have to illustrate the volume of tumor growth observed in FASN^{+/+}PyMT versus FASN Δ/Δ PyMT mice.

Figure 6B: Authors should illustrate higher magnifications for both illustrations. As authors write in lines 332, 333 that the number of foci is smaller in PyMT/FASN negative tumors compared to PyMT/FASN positive tumors, they have to quantify and illustrate such quantifications.

Figure S6D,E: Authors claim that similar differences are observed in vivo using MEFs models or HC cells (line 338) which is not correct. A decrease in tumor size is indeed observed following FASN deletion using MEFs cell lines with KRAS and HER2 mutations (FigS6B, C) however using HC11 breast cancer cell lines (Fig6 D,E) tumor growth is only delayed and not reduced as authors suggest by mentioning "similar differences were observed". This points out the fact that FASN deletion is efficient to impede growth of tumors derived from MEFs (following KRAS or HER2 mutations) but not of tumors derived from HC11 cells for which only a delay of growth is observed. To increase the relevance of targeting FASN in vivo, I would suggest that authors combine the model illustrated in Fig6 with G28UCM treatment to eradicate remaining FASN positive foci and evaluate the impact of a total deletion of FASN on spontaneous breast cancer tumors.

Figure 7 : Examination of ROS levels or ROS induced DNA damage should also be performed in FASN^{+/+}PyMT mice with or without G28UCM as in figure 6C.

Figure 6 and 7 : Is the FASN^{+/+}PyMT suitable to evaluate metastasis formation? If so authors have to evaluate and illustrate the metastatic index in FASN^{+/+}PyMT versus FASN Δ/Δ PyMT or with or without G28UCM.

Statistical analysis: Information about statistical analysis, biological replicates and experimental replicates are lacking in many figures. For all panels in figures authors should mention the exact number of experimental replicates, example : Fig2C, D, Fig 4B (and avoid sentences such as "at least 3 experiments"). In Fig1A: n=6 : is this biological replicates or experimental replicates?

Discussion : In practice, how authors consider the feasibility to clinically target FASN before transformation? In high risk patients? This concept needs to be clarified.

Minor comments:

Figure 3D: There is no compensatory induction of Glut1 as claimed by authors as it appears ns. Authors should correct this in the manuscript. Glut4 mRNA levels in FASN^{lox/lox}PyMT is below 1, what is the sample of reference as =1?

Figure S2B, S5A : Addition of glucose, oligomycin and 2DG graphs have to be corrected for ECAR panel. Authors should carefully check legends of their graphs.

Line 254 : "extra-mitochondrial" should be corrected by "intra-mitochondrial"

Figure 1c : axis legends of needs scale

Title of Figure 1 should be focused on clonogenic ability of MEFs as proliferation is not altered by FASN deletion.

Line 135 : Authors should add "and apoptotic rates were unaffected by FASN depletion in..."

Reviewer #2: FASN

(Remarks to the Author):

Bueno et al describe a new role for FASN, quenching excess ROS produced during anchorage independent transformation through consumption of acetyl-CoA and IDH-dependent reductive carboxylation. The authors show how FASN activity is essential for the initial steps of transformation using different oncogene driven models, and characterize the levels and fate of metabolic intermediates in cells but that this activity is necessary to restore the NAD⁺/NADH ratio and is independent of palmitate production. Overall, the study is original, interesting, and potentially relevant, but some of the claims are not supported by the data provided and additional experiments are required to substantiate the conclusions.

1. The most relevant and novel finding in this paper is the fact that FASN is essential to sustain IDH-1 dependent reductive carboxylation of glutamine. The rescue of spheroids and soft agar growth after restoring NAD⁺/NADH ratio is very interesting and important in support of the hypothesis. To demonstrate that this is indeed a universal phenomenon, the authors should include data in figure 4 (or figure S5) presenting both NAC/GSH rescue in all oncogene models studied (PyMT, Kras, HER2).

2. The authors claim that the effects of FASN on cell transformation is independent of its enzymatic product palmitate, stating that the preferential source of fatty acids for the cells is exogenous. This is a pretty strong claim that is counter to several studies demonstrating that the lack of de novo palmitate synthesis is the main reason for the induction of apoptosis, cell growth reduction, etc, typically observed following FASN inhibition. However, this study is carried out in fibroblasts transformed by oncogenes. Normal cells do not require FASN activity as they rely on exogenous lipids and the MEFs being utilized perhaps have yet to achieve the metabolic re-wiring required during full transformation. Requirements for endogenously produced lipids (palmitate) is different in neoplastic cells. In addition, the palmitate rescue experiment in MEFs (figure S3C) was not performed with albumin-complexed palmitate, which might have negatively affected the result. Another issue is in figure S1D: in the absence of exogenous palmitate no colony formation was observed in MEFs with any of the oncogene models evaluated. Palmitate represents 26% of fatty acid content in regular FBS, as stated by the authors. Therefore, fatty-acid free FBS is also depriving the cells of many other lipids, confounding results. Two approaches could be done to support that de novo synthesis of palmitate is not playing a role in colony formation: add exogenous albumin-complexed palmitate to fatty acid free FBS to rescue the colony formation capability, or add sulfo-N-succinimidyl oleate (an inhibitor of fatty acid uptake) to medium containing normal FBS, ensuring that the uptake of fatty acids is, indeed, necessary for cell transformation.

3. In figure S3C, soraphen-A was used to decrease malonyl-CoA availability, preventing fatty acid oxidation (FAO) blockade and, consequently, impairment of colony formation. While no growth rescue was observed, the authors did not verify FAO activity, or whether soraphen-A could rescue the potential FAO blockade observed with FASN ablation.

4. In figure 3B the unlabeled lipid pools were similar in FASN^{lox/lox}-PyMT and FASN^{Δ/Δ}-PyMT MEFs. The authors believe this result confirms that intracellular lipids levels are not dependent on de novo lipogenesis and that cell transformation is not dependent on palmitate itself, which again conflicts with other studies that show the critical role of FASN in critically contributing to the intracellular pool of lipids. Neutral lipid accumulation assessed by Oil Red O staining could provide further confirmation of this result.

5. The authors claim that IDH1-driven reductive carboxylation supports anchorage independence, which is blocked in the absence of FASN activity due to acetyl-CoA accumulation that leads to an increased citrate/isocitrate ratio. The carbon tracing experiments and the addition of sodium citrate or ATP citrate lyase inhibitor lend support for the mechanism proposed. However,

considering the critical role of IDH1 in this context, confirmation by genetic ablation of IDH1 would strengthen the data in figures 4 and 5.

6. Data from pharmacologic inhibition of FASN in preclinical models requires further investigation. Testing G28UCM compound in FASN Δ/Δ -PyMT mice would represent a necessary control for off-target effect. This could be performed in MEFs derived from FASN Δ/Δ -PyMT mice compared to FASN $^{lox/lox}$ -PyMT mice-derived MEFs. Also, since the level of FASN expression is high in both vehicle and G28UCM-treated tumors, it is necessary to determine FASN activity after treatment, radiolabeling acetyl-CoA/malonyl-CoA and analyzing the labeled lipid pool.

7. The authors suggest to target FASN before transformation, as a prevention strategy. Normal cells, however, do not require FASN activity as they rely on exogenous lipids. This is different in neoplastic cells. Thus, this suggestion lacks the biological rationale.

8. There are certainly other mechanisms that require FASN activity when cells acquire invasive properties and these are not mentioned or tested (PMID: 22238651).

Rebuttal Letter – NCOMMS-18-31933-T

Reviewers' comments:

Reviewer #1: Cancer metabolism (Remarks to the Author):

In this study, Bueno et al. show that FASN is required for cell transformation, is essential to maintain low ROS levels and thereby promotes anchorage independent growth. Interestingly, authors show that the role of FASN as a promoter of oncogenic transformation is not dependent on its synthetic products, but mainly relies on an impairment of IDH-1-dependent reductive carboxylation, previously described to be essential for 3D growth (Jiang et al., Nature, 2016). Indeed, authors conclude that FASN deletion induces acetyl-CoA accumulation which consequently inhibits ATP citrate lyase and thereby promotes citrate/isocitrate accumulation. This later accumulation limits IDH1-dependent reductive carboxylation observed in 3D growth and which is needed to supply mitochondria with reduced equivalent to quench excess of ROS produced in mitochondria following spheroids formation. Authors show robust in vitro metabolic tracing data and perform several in vivo models to validate the role of FASN as a promoter of tumor growth.

Major Comments :

1) Figure S1: Levels of FASN are increased in MEFs following PyMT infection. This increase is not observed following KRAS or HER. Could this increase of FASN participate to the ability of MEFs to transform upon PyMT? In other words do levels of FASN impact on transformation by PyMT?

Levels of RAS are higher in FASN^{lox/lox} MEFs compared to FASN^{+/+} MEFs. As FASN^{lox/lox} MEFs proliferate more (Fig1C) they probably are easier to infect and integrate more retroviral particules which could explain the increased RAS level. So one could conclude that increased RAS levels in FASN^{lox/lox} MEFs participate to the increased clonogenic ability of these cells (Fig S1C) and could be responsible to the rapid tumor growth at T3-T4 observed in FigS6B compared to the slower tumor growth at T8 for FASN^{+/+}KRAS⁺ MEFs. Quantification of western blots (Fig S1) would help.

These two are actually great points. It is known from clinical observations that FASN is upregulated in transformed epithelial malignancies compared to paired non-transformed tissues of the same lineage. It is expected that FASN will increase after an oncogenic insult (PyMT, or KRAS-HER2). Our fault was not to include examples of different transformations: we include now examples of 6 different transformations from PyMT, KRAS and HER2. It can be observed how FASN levels display minor fluctuations but not really a tendency of higher values in transformed versus non-transformed clones. These results are shown now in Supplementary Figures 2a, b and c.

These minor variations did not impact in the clonogenicity of transformed clones: Clones with different FASN levels carrying PyMT, KRAS or HER2 showed equally proficient colony formation capacity. These results are shown now in Supplementary Figures 2d, e

and f.

Regarding the replication rate in 2D and the integration of retroviral particles, again, our fault was not to include more examples. We show now that the levels of KRAS, PyMT or HER2 fluctuate among FASN^{lox/lox} and FASN^{Δ/Δ} clones. These results are included now in Supplementary Figure 2g.

These results were preserved *in vivo*. KRAS levels (IHC and WB) were similar in tumors arising from FASN^{lox/lox} and FASN^{Δ/Δ} clones (now shown in Supplementary Figures 7g and 7h). It is true that the replication rate observed *in vitro* (Figure 1c) could lead to the hypothesis that *in vivo* FASN^{Δ/Δ} clones form no tumors or do so at a delayed pace because of a slow replication, but, concordantly with what we have observed, the problem (*in vivo*) is not a decreased replication rate, but the inability to switch from 2D-growth to 3D-growth. FASN^{lox/lox} and FASN^{Δ/Δ} clones originated tumors that display similar Ki67 replicative staining. These results are shown in Supplementary Figure 7f.

We have included the following text (plus the novel figures and figure legends) at the end of the first epigraph of the results section regarding the *in vitro* queries (FASN and KRAS/PyMT/HER2 levels):

“Although FASN^{Δ/Δ} clones were unable to form colonies, it is important to mention that minor FASN level fluctuations observed among different FASN^{lox/lox}-PyMT clones were not related to an impaired or improved clonogenic ability (Supplementary Figures 2a, d); similar observations applied for FASN^{lox/lox}-KRAS and FASN^{lox/lox}-HER2 (Supplementary Figures 2b, c, e and f). Finally, although non-statistically significant, the lower replication rate of FASN^{Δ/Δ} clones in 2D cultures could account for a decreased integration of retroviral

particles containing the studied oncogenes, but the examination of PyMT, KRAS and HER2 levels across different FASN^{Δ/Δ} and FASN^{lox/lox} clones did not reveal significant differences (Supplementary Figure 2g).”

Regarding the *in vivo* data, we have included the following text in the last epigraph of the result section (in addition to the novel figures and figure legends):

“In line with the *in vitro* data (Supplementary Figure 2g), we did not observe different replication levels (Ki67 staining) or oncogene levels (KRAS, HER2) in tumors originated from FASN^{lox/lox} or FASN^{Δ/Δ} clones (Supplementary Figures 7f, g and h)”

2) Figure S1E: HC11FASNKO cell lines are generated by the Crispr-C9 technology. sgRNA targeting human FASN were used. It appears that HC11FASNwt cells were not infected by recombinant lentivirus containing control sg. Such control is missing. Authors should compare both lentivirus infected HC11 cells with or without FASN.

We regret not having included those controls. When we were executing the HC11 experiments we had everything else already finished, and we were just validating the results in one more system with further oncogenes. At that moment the phenotype was so consistent that we sincerely doubted that the Sg integration could play any role on itself, at least on this phenotype. Regardless, and because academically speaking it is the correct control, although we performed all the experiments with the HC11 “wild-type” clones, we

tested the role of the Sg integration as a potential interference by generating HC11 clones infected by the control Sg, in the key experiments [soft-agar colony formation (now shown in Supplementary Figures 1e, f), OCR, ECAR (now shown in Supplementary Figure 3b), spheroids formation (now shown in Supplementary Figure 5a) and ROS (now shown in Supplementary Figure 5b)]. In all cases, scrambled Sg clones showed the exact same phenotype than the WT cells.

In addition, we would like to mention that the reference to human FASN in the methods section was a mistake. It is murine FASN. It has been corrected.

3) Figure S2A: Authors mention that the decrease in PDH activity in FASN^{Δ/Δ}PyMT MEFs compared to FASN^{lox/lox}PyMT MEFs is phenocopied by adding acetylCoA to FASN^{lox/lox}PyMT MEFs which is not clearly illustrated in the panel S2A where no statistic are illustrated.

We thank the Reviewer for noticing this, since the previous version included only three comparisons. We have included a new version of the figure with the P value for the requested comparison of the PDH activity between FASN^{lox/lox}-PyMT and FASN^{Δ/Δ}-PyMT clones, which yielded a P value of 0.213 (T-test).

Supplementary Figure 2a is now Supplementary Figure 3a.

4) Figure 2B. NADPH production is increased in FASN^{Δ/Δ}PyMT MEFs compared with FASN^{lox/lox}PyMT MEFs in 2D condition. Is this increase due to low ROS levels in FASN^{Δ/Δ}PyMT MEFs or vice-versa? Authors mention that, in 3D, there would be insufficient intra-mitochondrial reduced

equivalents which are usually consumed by excess of unquenched ROS produced during the 2D to 3D transition. NADPH measurements in 3D should be performed and added in Figure 4 or 5.

Moreover, authors should better explain why accumulation of unquenched ROS during 2D to 3D transition would be followed by impaired mitochondrial respiration, as authors write that lack of ROS production in 2D in FASN^{Δ/Δ}/PyMT MEFs explains the low OCR observed (lines 165 to 168). These 2 different conclusions in 2D and 3D conditions are confusing. Collaboration with expert in ROS and OXPHOS mechanisms such as Dr N. Chandel, co-authors of this study, should help to make these points cleared up.

We thank the Reviewer from bringing up this point, since it is the “core” of the manuscript and was not well explained in the previous version.

In 2D, FASN^{Δ/Δ}-PyMT displays a low incorporation of pyruvate into the Krebs cycle and decreased Krebs cycle activity (Figure 3), which in turn implies low mitochondrial activity and respiration. This is associated to low ROS. The increased NADPH observed in 2D in FASN^{Δ/Δ}-PyMT can be explained by the low ROS, but, above all, by the simple fact that FASN synthesizes fatty acids by condensing acetyl CoA, malonyl CoA and NADPH. If FASN is not working, there is not NADPH consumption by FASN, and FASN is one of the main consumers of NADPH (Figures 2b and f). In contrast, in 3D (where cells generate more ROS according to already published observations¹; we observed the same effect), the main contributor to low mitochondrial activity is the excess of ROS generated by the citrate lyase blockade. NADPH becomes key for tumor progression in 3D, since, wild type clones

quench the excess of ROS by NADPH originated through reductive carboxylation; the knockouts can not produce these extra NADPH, although the amount of NADPH would never be “0” because FASN is inactive and it is not consuming NADPH at all. Thus, it would be expected to observe increased NADPH in the knockouts in 2D (already in previous version) but decreased NADPH in 3D.

How can we prove that the chain of events is as we say, and that it is the excess of ROS what negatively impacts mitochondrial respiration in 3D in the knockouts? By two experiments that are now included in the manuscript:

-First, we have already showed low Krebs cycle activity in 2D, low ROS and high NADPH. In 3D, we have shown high ROS, but not NADPH. We include now total and intra-mitochondrial NADPH in 3D (FASN^{lox/lox}-PyMT and FASN^{Δ/Δ}-PyMT clones), which show the opposite pattern: in 3D, NADPH (total and mitochondrial) is lower in FASN^{Δ/Δ}-PyMT clones. These data are now shown in Figure 5g.

-Second, and more important: In the previous version we have assessed the mitochondrial function (Seahorse, ROS, etc), but not the levels of mitochondrial supercomplexes. Since it has been reported that high ROS can disrupt the integrity of the complexes and cease respiration, we have studied the proportion of complex I assembly into supercomplexes. In FASN^{Δ/Δ}-PyMT clones in 2D, mitochondrial supercomplex I is intact and functional; they simply have low input (low Krebs activity). However, in 3D, mitochondrial complex I assembly into supercomplexes is much less abundant. The question of whether high ROS are due to an increased respiration, or, conversely, are the cause of a stalled respiration, can only be answered by this experiment: in our case, we have high ROS and dis-assembled complex I, what it is only compatible with high (unquenched) ROS causing mitochondrial respiration cease. The complex I assembly into supercomplexes in FASN^{lox/lox}-PyMT

clones is intact in all conditions. These data are shown in Figure 5h.

Previous Figures 5g, 5h, 5i and 5j are now 5i, 5j, 5k and 5l.

Thus, the model is the following:

-2D: background ROS production conditions. FASN^{lox/lox}-PyMT clones are showing standard respiration and ROS production. FASN^{Δ/Δ}-PyMT clones, however, have a decrease input in the Krebs cycle and thus produce less ROS. Both clones have intact mitochondrial electron transport chain complexes. NADPH is not required to quench 3D-derived ROS, and thus it is not unexpected to observe increased NADPH in FASN^{Δ/Δ}-PyMT clones, since the mitochondrial ROS production is low, and NADPH is not used for fatty acid synthesis.

-3D: increased ROS production conditions. FASN^{lox/lox}-PyMT clones can quench them by the excess NADPH produced by reductive carboxylation, and thus preserve complexes integrity and replicate/grow normally. FASN^{Δ/Δ}-PyMT clones, however, have a decreased ability to perform reductive carboxylation, evidenced by the data in the previous version, and the lower NADPH levels, which results in increased unquenched ROS that damage the mitochondria. In this situation, they can not sustain 3D growth and thus tumor formation.

The title of the results section has been changed to the following:

“Cells lacking FASN cannot grow in an anchorage-independent manner in 3D due to insufficient ROS quenching, leading to mitochondrial complex I disassembly from supercomplexes and explaining the inability to complete oncogenic transformation”

The text of the final part of the section is now as follows:

To test this hypothesis, first we measured intra-mitochondrial and total NADPH. Total NADPH levels were decreased in FASN^{ΔΔ}-PyMT compared to FASN^{lox/lox}-PyMT MEFs (Figure 5g). However, the difference was more pronounced in the intra-mitochondrial compartment (Figure 5g), which is congruent with the fact that cytoplasmic levels may still be high because of the lack of activity of FASN, which consumes NADPH. It has been shown that an increased ROS production could disrupt mitochondrial complexes integrity²,³, stalling respiration. While in 2D the decreased respiration levels of FASN^{ΔΔ}-PyMT can be attributed to a decreased entrance of pyruvate in the mitochondria, in 3D the increased ROS might be a more important contributor. In fact, we observed that in 2D total NADPH was increased in FASN^{ΔΔ}-PyMT clones (Figure 2b); however, in 3D in the context of decreased citrate lyase activity, we observed decreased intramitochondrial NADPH levels, increased ROS and decreased levels of supercomplex I (Figure 5h). In order to ascertain the functional impact of these observations, we cultured FASN^{ΔΔ}-PyMT MEFs in the presence of either NAC or GSH-MEE, two ROS-quenching agents. This approach rescued OCR (Figure 5i), tumor spheroid formation, and soft agar colony formation (Figure 5j) while reducing ROS burden in 3D (Figure 5k). The NAD⁺/NADH quotient was also restored after addition of NAC/GSH-MEE (Figure 5l). We observed similar rescues in the other studied FASN-negative systems expressing different oncogenes (Supplementary Figures 6 a, b). Taken together, our data suggest the acetyl-CoA build-up secondary to FASN deletion inhibits ATP citrate lyase and induces an accumulation of citrate/isocitrate. Subsequently, this impairs IDH1-dependent reductive carboxylation, a tumor cell requirement limited to 3D growth, inducing a decrease in intramitochondrial NADPH, an increase in ROS, and finally mitochondrial supercomplexes assembly disruption, which ultimately impairs cell

transformation and tumorigenesis.

Finally, the second-last paragraph of the discussion has been modified accordingly:

“... impaired, what leads to the observed decreased in intra-mitochondrial NADPH (Figure 5g) and decreased levels of mitochondrial complex I assembly into supercomplexes (Figure 5h), which accounts for the inability to transform (Figures 4 and 5). This was proven by carbon tracing, enzymatic activity determination, ROS levels, mitochondrial complexes integrity analysis and phenocopying/phenotype rescuing experiments with several metabolites and inhibitors across several different genotypes. Excessive ROS....”

5) Figure 3E Authors should illustrate other TCA intermediates such as fumarate, malate.

Several additional TCA intermediates are shown now in Figure 3e (figure legend corrected accordingly).

6) Figure 4: Authors should examine in MEFs if chemical FASN inhibitor recapitulates some of the features of the FASN^{-/-}PyMT phenotype in several 3D assays in order to strengthen their in vivo data using the G28UCM compound.

This is a very good, albeit previously missing, point. We have studied the effects of G28UCM administered to FASN^{-lox/lox}-PyMT MEFs in terms of spheroid formation, ROS production in 3D, and OCR. G28UCM disrupts spheroid formation, increases ROS production in 3D, and decreases OCR rates, although in a less intense manner than the genetic experiments.

The results are included (because of the high number of experiments already shown in Figure 4, and because of the text flow) in Supplementary Figure 8.

In the main text, when G28UCM is introduced, we have added an additional sentence:

“*In vitro* (Supplementary Figure 8), G28UCM recapitulated the main features of FASN deletion.”

7) Figure 6: Authors have to illustrate the volume of tumor growth observed in FASN^{+/+}PyMT versus FASN^{-/-}PyMT mice.

We have added a new panel (Figure 6f) including the evolution of tumor volume along time, and also three charts plotting tumor burden at three representative timepoints (Weeks 4, 5 and 6). The main text and figure legend have been modified accordingly.

8) Figure 6B: Authors should illustrate higher magnifications for both illustrations. As authors write in lines 332, 333 that the number of foci is smaller in PyMT/FASN negative tumors compared to PyMT/FASN positive tumors, they have to quantify and illustrate such quantifications.

We have added new pictures (high and low magnification, Figure 6b) that illustrate the point. However, we are afraid that quantifications are not possible (or at least not informative). The new PYMT/FASN positive picture shows a whole mounted tumor that illustrates how “everything is one tumor”. It is impossible to count such phenotype as number of foci. There are not non-tumor areas. Conversely, if we take a look at the

PyMT/FASN negative mounts, there are several (many) FASN-positive foci. It is clear that the tumor initiates in less places than in WT animals, but it is not possible to come up with a realistic number

9) Figure S6D,E: Authors claim that similar differences are observed in vivo using MEFs models or HC cells (line 338) which is not correct. A decrease in tumor size is indeed observed following FASN deletion using MEFs cell lines with KRAS and HER2 mutations (FigS6B, C) however using HC11 breast cancer cell lines (Fig6 D,E) tumor growth is only delayed and not reduced as authors suggest by mentioning “similar differences were observed”. This points out the fact that FASN deletion is efficient to impede growth of tumors derived from MEFs (following KRAS or HER2 mutations) but not of tumors derived from HC11 cells for which only a delay of growth is observed. To increase the relevance of targeting FASN in vivo, I would suggest that authors combine the model illustrated in Fig6 with G28UCM treatment to eradicate remaining FASN positive foci and evaluate the impact of a total deletion of FASN on spontaneous breast cancer tumors.

The reviewer is absolutely right, we apologize for having not been accurate in our wording.

We have change the wording – the new text is as follows:

“We tested as well the *in vivo* effects on tumor growth of FASN^{Δ/Δ}-MEFs infected with KRAS or HER2, grafting them into wild-type animals compared with FASN^{lox/lox} counterparts (Supplementary Figures 7b, c). In line with the *in vitro* data (Supplementary Figure 2g), we did not observe different replication levels (Ki67 staining) (Supplementary

Figure 7f) or oncogene levels (KRAS, HER2) in tumors originated from FASN^{lox/lox} or FASN^{Δ/Δ} clones (Supplementary Figures 7g, h). A strong reduction in tumor growth was observed (Supplementary Figures 7b, c). Finally, when HC11 cells with CRISPR-deleted FASN were infected with PyMT or KRAS and grafted into wild-type animals, a delay on tumor onset was observed compared with wild-type counterparts (Supplementary Figures 7d, e).”

The combination of a pharmacologic and genetic approach is an excellent idea, and we have performed the experiment. We have observed a further decrease in tumor burden in the FASN-negative PyMT animals when they received G28UCM treatment compared with vehicle. However, the treatment was toxic in these animals (likely because of the complete or almost-complete systemic FASN function ablation; the animals experienced diarrhea and weight loss) and most animals only reached timepoint +3 weeks. The data are shown in Supplementary Figure 8.

10) Figure 7: Examination of ROS levels or ROS induced DNA damage should also be performed in FASN^{+/+}PyMT mice with or without G28UCM as in figure 6C.

We thank again the Reviewer for a great suggestion to make the mechanistic findings more robust.

We have performed the experiment, and found that G28UCM induces high ROS levels in FASN^{+/+}; PyMT animals. The results are shown in Supplementary Figure 9. We have added the following sentence in the main text:

“In addition, in agreement with the proposed mechanism of action of FASN deletion, when G28UCM was applied to FASN^{+/+}; PyMT animals, their tumors displayed intense 8-oxo-dG staining (Supplementary Figure 9).”

11) Figure 6 and 7: Is the FASN^{+/+}PyMT suitable to evaluate metastasis formation? If so authors have to evaluate and illustrate the metastatic index in FASN^{+/+}PyMT versus FASN^{-/-}PyMT or with or without G28UCM.

This is a great point that actually sets the grounds for future work and likely a full manuscript. Usually, studies on the PyMT model rely on mixed background or pure C57/B6 background. In these backgrounds, tumor onset starts around 16-20 weeks, and animals live up until 30-35 weeks (when tumors reach the humane endpoint). The long time during which the animals host the tumors allows for metastases formation, and the usually reported metastatic rate (in the lungs) is around 50-100% (in our hands – with our strain – it is closer to 50%).

We have backcrossed the PyMT animal to a pure FVB background and to a pure C57/B6 background. We use the latter for studying the function of novel oncogenes or tumor promoters/accelerators, while we use the former for studies with genes involved in a potential delay or abrogation of tumor onset or growth (the model is also very good for oncologic drugs, since due to the fast growth it allows studying antitumor effects of novel compounds relatively fast).

Since the pure FVB-PyMT animals usually have to be sacrificed at 5-7 weeks after tumor onset, the metastatic ratio is very low – approximately 10% in our hands. Thus, it is an impractical model for answering this question. We have examined the lungs from 54 FVB-PyMt animals sacrificed at the humane endpoint, and we have found metastases in 5

animals (9.5% of the cases).

In addition, the following picture allows appreciating that the metastatic disease burden is quite low.

In summary, it is a great question that we also want to address. We believe that because FASN^{-/-} cancer cells are unable to survive in 3D, we believe that they will be unable as well to form metastases because of the intermediate steps required for doing so (surviving in a cell-adhesion independent manner in blood, forming CTC-clumps, give rise to a 3D tumor in the lung, etc). The study of the process will require heavy experimentation and mechanistic studies, and also move to the C57/B6 background. Thus, it will be reported in a future manuscript in about 2-3 years from now.

12) Statistical analysis: Information about statistical analysis, biological replicates and experimental replicates are lacking in many figures. For all panels in figures authors should mention the exact number of experimental replicates, example : Fig2C, D, Fig 4B (and avoid sentences such as “at least

3 experiments”). In Fig1A: n=6 : is this biological replicates or experimental replicates?

The number of replicates (and whether they are biological or technical), as well as information about the statistical test that was performed, has been incorporated in all principal and supplementary figures wherever it applies.

13) Discussion: In practice, how authors consider the feasibility to clinically target FASN before transformation? In high risk patients? This concept needs to be clarified.

We thank the reviewer for giving us the opportunity to clarify this point. Indeed, akin any other intervention in the prevention sphere involving drugs and not just lifestyle changes, it would be justified only in high-risk patients (at least until some benefit is demonstrated; subsequently, further trials in lower-risk or standard-risk populations can be planned).

Several trials have assessed various approaches for decreasing the rate of invasive cancer^{4,5}, and usually they rely on the same criteria: Gail score above 1.7 (in 5 years), previous history of lobular carcinoma in situ and/or previous history of atypical hyperplasia.

We have modified the last paragraph of the manuscript accordingly:

“Taken together, these features indicate FASN is a potential target for cancer prevention. Akin previous chemoprevention trials^{4,5}, future interventions should be planned in healthy patients at high risk of developing invasive breast cancer, such as those with previous

diagnosis of atypical hyperplasia, lobular carcinoma in situ, or high Gail 5-year risk score.”

Minor comments:

14) Figure 3D: There is no compensatory induction of Glut1 as claimed by authors as it appears ns. Authors should correct this in the manuscript. Glut4 mRNA levels in FASN^{lox/lox}PyMT is below 1, what is the sample of reference as =1?

We thank the Reviewer for noticing this. Regarding Glut 1, we have corrected the main text. In the case of Glut4, it was incorrectly plotted: the reference (FASN^{lox/lox}-PyMT in both charts) is 1 (Figure 3d).

15) Figure S2B, S5A : Addition of glucose, oligomycin and 2DG graphs have to be corrected for ECAR panel. Authors should carefully check legends of their graphs.

We apologize for the confusion – rather than a mislabeling it was the ECAR readout of the OCR experiment: i.e., in the previous version of the figure, since the Seahorse machine can measure simultaneously the oxygen consumption and the extracellular acidification rate, we provided the acidification rate in the experiment designed to measure OCR (that is, during the timecourse of adding oligomycin, FCCP and antimycin/rotenone). It is common to observe such type of simultaneous report in publications, however, it does not provide an accurate assessment of glycolysis. Thus, we have chosen to repeat the experiment, this time measuring ECAR with the ECAR-measuring protocol (i.e., sequential addition of glucose,

oligomycin and 2-DG).

Supplementary Figure 2 and Supplementary Figure 5 are now Supplementary figures 3 and 6, respectively. The new, corrected, ECAR panels are shown in those figures.

16) Line 254: “extra-mitochondrial” should be corrected by “intra-mitochondrial”

We thank the Reviewer for noticing this typo. It has been corrected.

17) Figure 1c: axis legends of needs scale

We are not 100% certain about the meaning of this question – in any case, we have tried our best to clarify the chart. The chart has two axes, X and Y, and plots the relative increase in the number of cells (Y axis) along time (X axis). Time is expressed in days (0 to 3). The Y axis is now re-labeled (“Fold-increase in cell number”), and expresses the relative number of cells of each clone at any given time compared to time 0, when the experiment started. We have clarified as well the figure legend: “Lack of significant differences in cell replication rates along time”.

18) Title of Figure 1 should be focused on clonogenic ability of MEFs as proliferation is not altered by FASN deletion.

We have changed the title of Figure 1 accordingly.

19) Line 135 : Authors should add “and apoptotic rates were unaffected by FASN depletion in...”

We thank the Reviewer for the suggestion, which has been incorporated.

Reviewer #2: FASN

(Remarks to the Author):

Bueno et al describe a new role for FASN, quenching excess ROS produced during anchorage independent transformation through consumption of acetyl-CoA and IDH-dependent reductive carboxylation. The authors show how FASN activity is essential for the initial steps of transformation using different oncogene driven models, and characterize the levels and fate of metabolic intermediates in cells but that this activity is necessary to restore the NAD⁺/NADH ratio and is independent of palmitate production. Overall, the study is original, interesting, and potentially relevant, but some of the claims are not supported by the data provided and additional experiments are required to substantiate the conclusions.

1. The most relevant and novel finding in this paper is the fact that FASN is

essential to sustain IDH-1 dependent reductive carboxylation of glutamine. The rescue of spheroids and soft agar growth after restoring NAD⁺/NADH ratio is very interesting and important in support of the hypothesis. To demonstrate that this is indeed a universal phenomenon, the authors should include data in figure 4 (or figure S5) presenting both NAC/GSH rescue in all oncogene models studied (PyMT, Kras, HER2).

We apologize for not having included all the possible variants. We include now proof of spheroid formation rescue on the missing cell lines/genotypes. As it can be appreciated (now Supplementary Figure 6), the results are consistent across different genotypes.

2. The authors claim that the effects of FASN on cell transformation is independent of its enzymatic product palmitate, stating that the preferential source of fatty acids for the cells is exogenous. This is a pretty strong claim that is counter to several studies demonstrating that the lack of de novo palmitate synthesis is the main reason for the induction of apoptosis, cell growth reduction, etc, typically observed following FASN inhibition. However, this study is carried out in fibroblasts transformed by oncogenes. Normal cells do not require FASN activity as they rely on exogenous lipids and the MEFs being utilized perhaps have yet to achieve the metabolic re-wiring required during full transformation. Requirements for endogenously produced lipids (palmitate) is different in neoplastic cells. In addition, the palmitate rescue experiment in MEFs (figure S3C) was not performed with

albumin-complexed palmitate, which might have negatively affected the result. Another issue is in figure S1D: in the absence of exogenous palmitate no colony formation was observed in MEFs with any of the oncogene models evaluated. Palmitate represents 26% of fatty acid content in regular FBS, as stated by the authors. Therefore, fatty-acid free FBS is also depriving the cells of many other lipids, confounding results. Two approaches could be done to support that de novo synthesis of palmitate is not playing a role in colony formation: add exogenous albumin-complexed palmitate to fatty acid free FBS to rescue the colony formation capability, or add sulfo-N-succinimidyl oleate (an inhibitor of fatty acid uptake) to medium containing normal FBS, ensuring that the uptake of fatty acids is, indeed, necessary for cell transformation.

We sincerely thank these suggestions, since we are aware that the commonly accepted notion is that the synthetic product itself is the essential feature linked to FASN upregulation in cancer. We were surprised as well to observe our results, and the suggestions made by the Reviewer strengthen even more the conclusion, which we believe that it is a major finding. We have performed both experiments, and, to our surprise, neither the MEFS (PyMT, KRAS or HER2) growing in full medium with sulfo-N-succinimidyl oleate (SSO), nor the MEFS (PyMT, KRAS, HER2) growing in DCC supplemented with albumin-complexed palmitate were able to form colonies. The data are shown in Supplementary Figure 4f. In addition (next query), we have observed that exogenous fatty acids are used for FAO (Supplementary Figure 4d).

We have modified the text in the results section accordingly:

“ The role of exogenous fatty acid in colony formation essentiality was further tested: First, palmitate represents only 26% of FBS fatty acid pools⁶; thus, other exogenous lipids could be essential for transformation. When FASN^{lox/lox}-PyMT, -HER2 or KRAS MEFs were incubated in full medium with FBS and the fatty acid uptake inhibitor sulfo-N-succinimidyl oleate (SSO)⁷, no colonies were recovered. Second, when fatty acid-free media was supplemented with albumin-complexed palmitate in order to facilitate its uptake by cancer cells, FASN^{lox/lox}-PyMT, -HER2 or KRAS MEFs did not form colonies in soft agar (Supplementary Figure 4f). Lastly, tumor epithelial cells from the breast of PyMT animals with (FASN^{+/+}; PyMT) or without (FASN^{Δ/Δ}; PyMT) FASN displayed similar intracellular neutral lipid accumulation evidenced by oil-red staining (Supplementary Figure 4g). These results indicate that fatty acid requirements of cancer cells are mainly satisfied by the uptake from free fatty acids, implying that FASN is required for transformation for a different reason than its synthetic product”

Of note, one part of the previous text (“Lastly, tumor epithelial cells from the breast of PyMT animals with (FASN^{+/+}; PyMT) or without (FASN^{Δ/Δ}; PyMT) FASN displayed similar intracellular neutral lipid accumulation evidenced by oil-red staining (Supplementary Figure 4g.”) corresponds to the answer of query #4 (See below)

In addition, we have modified the discussion section relevant to these data. The text is now as follows:

“The product-independent role of FASN is further supported by the following facts: 1) in 2D conditions, FASN^{Δ/Δ}-PyMT MEFs grow normally (Figure 1c) but not in 3D conditions

(Figure 4a); 2) palmitate or FFA-free-albumin-complexed palmitate do not rescue tumorigenesis in FASN^{ΔΔ}-PyMT MEFs (Supplementary Figure 4c); 3) the preferential source of fatty acids appears external, and maybe relying on fatty acids different from palmitate, evidenced by FASN^{lox/lox}-PyMT MEFs with intact FASN levels forming colonies only in the presence of fatty acids in the media, but losing the ability to form colonies in presence of SSO (Supplementary Figure 4f). Data from the measurement of unlabeled lipid pools in both genotypes (Figure 3b) and neutral lipid accumulation in the tumor epithelium of PyMT animals (Supplementary Figure 4g) further support this conclusion”

Finally, with the permission of the Reviewer, we have changed the wording regarding the relevance of the finding (from “Another relevant finding of our study” to “A very important finding of our study”).

3. In figure S3C, soraphen-A was used to decrease malonyl-CoA availability, preventing fatty acid oxidation (FAO) blockade and, consequently, impairment of colony formation. While no growth rescue was observed, the authors did not verify FAO activity, or whether soraphen-A could rescue the potential FAO blockade observed with FASN ablation.

We thank the Reviewer for pointing this out, since it has been proposed that one main mechanism by which FASN inhibition impairs tumor growth is because of the Malonyl-Coa build-up which in turn would inhibit fatty acid oxidation. We have found that the other FASN substrate – Acetyl-CoA – buildup prevents oxidation of glucose through the blockade of pyruvate entry in the mitochondria, but we have not measured FAO in the

previous version.

As expected, and as the Reviewer suggests, through the determination of labeled ^{14}C – CO_2 emission, we have detected that WT and FASN KO MEFs incorporate and oxidize exogenous palmitate. In addition, the KOs do so in a decreased ratio compared to the Wt counterparts. This decrease is rescued, as expected, by soraphen A (Supplementary Figure 4e). As shown in the previous version, treatment with soraphen A does not rescue colony formation (Supplementary Figure 4c). Thus, taken together, these results suggest that although the malonyl-CoA-buildup account for a decrease in FAO, this is not the ultimate cause of lack of ability to transform.

These results are shown in Supplementary Figures 4d-e.

We have modified the text accordingly in the third sub-section of the results section:

“Interestingly, in the presence of palmitate, neither soraphen-A nor dNTPs could rescue the transforming ability of PyMT in $\text{FASN}^{\Delta/\Delta}$ MEFs (Supplementary Figure 4c). FAO of exogenous, ^{14}C -labelled fatty acids, show that both $\text{FASN}^{\text{lox/lox}}$ -PyMT and $\text{FASN}^{\Delta/\Delta}$ -PyMT MEFs oxidize fatty acids, although $\text{FASN}^{\Delta/\Delta}$ -PyMT MEFs do so – as expected – at a lower rate (Supplementary Figure 4d). The addition of soraphen-A, as predicted, boosted and rescued, respectively, FAO in $\text{FASN}^{\text{lox/lox}}$ -PyMT and $\text{FASN}^{\Delta/\Delta}$ -PyMT MEFs (Supplementary Figure 4e), suggesting that although the malonyl-CoA buildup accounts for a decrease in FAO, this is not the reason why $\text{FASN}^{\Delta/\Delta}$ -PyMT can not undergo transformation.”

4. In figure 3B the unlabeled lipid pools were similar in $\text{FASN}^{\text{lox/lox}}$ -PyMT

and FASN Δ/Δ -PyMT MEFs. The authors believe this result confirms that intracellular lipids levels are not dependent on de novo lipogenesis and that cell transformation is not dependent on palmitate itself, which again conflicts with other studies that show the critical role of FASN in critically contributing to the intracellular pool of lipids. Neutral lipid accumulation assessed by Oil Red O staining could provide further confirmation of this result.

We thank the reviewer for this interesting suggestion.

First, when describing Figure 3b, we have “toned-down” the statement, and replaced “confirms” by “suggests”, since most of the experiments supporting the conclusion of Independence of de novo lipogenesis are explained in subsequent sections.

Second, in the paragraph where we explain the experiments with SSO and albumin-complexed palmitate, we have added the requested staining with Oil Red. Both the tumor epithelium from FASN^{+/+} animals and from FASN ^{Δ/Δ} animals show positive staining, suggesting, and further strengthening that the exogenous lipid pool represent a relevant fraction of intracellular lipid deposits. The results are shown in Supplementary Figure 4g.

The text in the results section and in the discussion section is as follows (it is the same text as the one pasted in query #2):

Results:

“The role of exogenous fatty acid in colony formation essentiality was further tested: First, palmitate represents only 26% of FBS fatty acid pools⁶; thus, other exogenous lipids could be essential for transformation. When FASN^{lox/lox}-PyMT, -HER2 or -KRAS MEFs were incubated in full medium with FBS and the fatty acid uptake inhibitor sulfo-N-succinimidyl

oleate (SSO)⁷, no colonies were recovered. Second, when fatty acid-free media was supplemented with FFA-free-albumin-complexed palmitate in order to facilitate its uptake by cancer cells, FASN^{lox/lox}-PyMT, -HER2 or -KRAS MEFs did not form colonies in soft agar (Supplementary Figure 4f). Lastly, tumor epithelial cells from the breast of PyMT animals with (FASN^{+/+}; PyMT) or without (FASN^{Δ/Δ}; PyMT) FASN displayed similar intracellular neutral lipid accumulation evidenced by oil-red staining (Supplementary Figure 4g). These results indicate that fatty acid requirements of cancer cells are mainly satisfied by the uptake from free fatty acids, implying that FASN is required for transformation for a different reason than its synthetic product.”

Conclusion:

“The product-independent role of FASN is further supported by the following facts: 1) in 2D conditions, FASN^{Δ/Δ}-PyMT MEFs grow normally (Figure 1c) but not in 3D conditions (Figure 4a); 2) palmitate or albumin-complexed palmitate do not rescue tumorigenesis in FASN^{Δ/Δ}-PyMT MEFs (Supplementary Figure 4c); 3) the preferential source of fatty acids appears external, and maybe relying on fatty acids different from palmitate, evidenced by FASN^{lox/lox}-PyMT MEFs with intact FASN levels forming colonies only in the presence of fatty acids in the media, but losing the ability to form colonies in presence of SSO (Supplementary Figure 4f). Data from the measurement of unlabeled lipid pools in both genotypes (Figure 3b) and neutral lipid accumulation in the tumor epithelium of PyMT animals (Supplementary Figure 4g) further support this conclusion”

5. The authors claim that IDH1-driven reductive carboxylation supports anchorage independence, which is blocked in the absence of FASN activity due to acetyl-CoA accumulation that leads to an increased citrate/isocitrate ratio. The carbon tracing experiments and the addition of sodium citrate or ATP citrate lyase inhibitor lend support for the mechanism proposed. However, considering the critical role of IDH1 in this context, confirmation by genetic ablation of IDH1 would strengthen the data in figures 4 and 5.

Once again, we sincerely thank the Reviewer for a suggestion that allows us to strengthen the conclusions.

We have silenced IDH1 with siRNA (we chose this approach to observe the effects of short-term IDH1 inhibition akin with a compound – SB204990 – or with citrate, since CRISPR editing might cause some sort of metabolic rewiring that would mask elucidating the mechanism) and we observed that it phenocopies the three main features of FASN deletion in 3D: increased ROS production, abrogation of formation of spheroids, and decreased OCR. The results are shown in Supplementary Figure 10.

We have added two lines of text, and a new supplementary figure, reporting the experiment in the main text:

“..... Figures 4i, j, Supplementary Figures 5d, e). When IDH1 was silenced with siRNA in 3D, it led to a disruption in the formation of tumor spheroids and suppression of OCR together with increased ROS, compared with no effect in these traits in 2D (Supplementary Figure 10). Taken together, ...”

6. Data from pharmacologic inhibition of FASN in preclinical models requires further investigation. Testing G28UCM compound in FASN Δ/Δ -PyMT mice would represent a necessary control for off-target effect. This could be performed in MEFs derived from FASN Δ/Δ -PyMT mice compared to FASN $^{lox/lox}$ -PyMT mice-derived MEFs. Also, since the level of FASN expression is high in both vehicle and G28UCM-treated tumors, it is necessary to determine FASN activity after treatment, radiolabeling acetyl-CoA/malonyl-CoA and analyzing the labeled lipid pool.

Determining on-target and off-target effects are important points on studies reporting on new drugs. Although our study is not interested in G28UCM (a compound that has been previously characterized⁸) and our only interest on it was as a tool compound for completing some of the experiments, we think that it is a good suggestion to wrap-up our findings in a purist academic manner.

The question has two parts – showing the on-target effect and controlling off-target effects. The first one is easy to demonstrate, while the other poses problems (see below).

On target effects: we tested fatty acid radioactive labeling after incubation with ^{14}C -acetate in FASN $^{lox/lox}$ -PyMT and FASN Δ/Δ -PyMT MEFs, in presence or absence of G28UCM. In FASN $^{lox/lox}$ -PyMT MEFs, G28UCM decreased approximately 40% the incorporation of ^{14}C -acetate, which was obviously less than the effects induced by FASN deletion (which is not 100% efficient, explaining the residual incorporation observed in FASN Δ/Δ -PyMT MEFs). When G28UCM was added to FASN Δ/Δ -PyMT MEFs, no significant differences were observed. These data are shown in Supplementary Figure 8d. In the main text, we have added the following statement:

“In vitro, G28UCM recapitulated the main features of FASN deletion and showed moderate on-target effect (Supplementary Figures 8a-d)”.

Regarding the off-target effects: it is common to have this request in manuscripts dealing with “targeted therapies”, in order to prove that a drug does have or does not have effects in absence of the target, after knocking it out. In the first case, the drug would have substantial off-target effects, and in the second, it would be relatively clean. As reviewers wisely usually ask for it, it is common now to assume the fact that most “targeted drugs” are not as “targeted” as they initially look, and thus is a legit question in order to characterize compounds.

In order to address such problem, two features must be taken into account: first, there has to be a phenotype (most commonly is tumor cell death), and second, there must be a pharmacodynamic surrogate. In a hypothetical example we could talk, for example, about a MEK inhibitor that has anticancer effect. We would observe certain cell killing (e.g., 70%), and we would observe pharmacodynamic correlate (i.e., in this case, decrease in pERK). In order to assess whether our MEK inhibitor has or not off target effect, we would KO MEK, and treat the Kos with the drug. Cells would be alive in absence of the drug (since our target would not be essential), and if our compound has off target effect, still a percentage of MEK KO cells would die with the compound - i.e., 20%, - but no further changes in pERK should be seen. Thus, the efficacy of our compound would be partly because of its effects in MEK (rough numbers 50%), and because of other off-target effects (roughly 20%).

Now, if we translate this to our scenario, there is a technical hurdle difficult to overcome: our phenotype is “abrogation of 2D to 3D transition in absence of FASN”. When we KO

FASN, the penetrance of the phenotype is 100%. We have never been able to recover colonies in soft agar or espheroids in low adherence when FASN is knocked out before PYMT transformation (with the exception of escapers, but that's a different story, since they still are FASN+). In other words, there's no "residual" phenotype where we can check with G28UCM if there's further decrease in "abrogation of 2D to 3D transition in absence of FASN, mediated through an effect through a different target than FASN" (thus proving that there's some other pathways that abrogate this transition, outside FASN-IDH1-reductive carboxylation) adding G28UCM to the knockouts and compare it to the knockouts alone, because the abrogation is already complete. In fact, when we add the compound in FASN wild type MEFs, what we observe is that it only partially abrogates 2D to 3D, because it only inhibits FASN partially. Unfortunately, there is no way to address whether in absence of FASN other G28UCM-modulated pathways come into play, because there's nothing left to block with G28UCM.

Binding studies for sure would find that G28UCM binds other targets, since the amount of compound required to inhibit FASN is relatively high; however, as we said, the manuscript is not reporting on G28UCM characterization.

7. The authors suggest to target FASN before transformation, as a prevention strategy. Normal cells, however, do not require FASN activity as they rely on exogenous lipids. This is different in neoplastic cells. Thus, this suggestion lacks the biological rationale.

We are not 100% sure we understand the question. What we meant is exactly what the Reviewer explains, which to the best of our understanding is the basis of

prevention: using a compound that would be relatively innocuous to normal cells (Rev: "Normal cells, however, do not require FASN activity") that would, meanwhile, impair the fitness of cells that are undergoing transformation (Rev: "This is different in neoplastic cells"). Such compound could be administered indefinitely, and, whenever a malignant transformation event occurs, the compound would at least in some individuals abrogate or delay its effects. Since today it is – yet – impossible to predict or detect when/where a malignant transformation events takes/will take place, that is the basis of chemoprevention: in groups of patients at high-risk (in the case of breast cancer, BRCA mutants or patients with high Gail score), they would be taking the compound while there is no evidence of disease, so that in case in one patient an early tumor has started to grow, or will start to do so, is blocked by the compound with selective activity in blocking the initial steps of tumorigenesis, which is what a clinical-grade compound would do. We believe that targeting specific processes of malignant cells or cells undergoing transformation while sparing normal cells is a quite reasonable biologic rationale for oncology therapeutics, regardless whether it is aimed at the prevention or therapeutic phase, and this rationale is behind the immense majority of existing anticancer drugs.

We have changed the wording of the prevention paragraph in any case to facilitate the reading.

" A clinical-grade compound that would selectively target FASN could be administered long-term to a healthy individual, sparing toxicity to normal tissues, while could target cells in their initial steps of transformation. Taken together, these features indicate FASN is a potential target for cancer prevention. Akin previous chemoprevention trials^{4, 5}, future interventions should be planned in healthy patients at high risk of developing invasive

breast cancer, such as those with previous diagnosis of atypical hyperplasia, lobular carcinoma in situ, or high Gail 5-year risk score.”

8. There are certainly other mechanisms that require FASN activity when cells acquire invasive properties and these are not mentioned or tested (PMID: 22238651).

We have added the following sentence introducing the last paragraph of the discussion so that the reader can put our findings in context, citing the manuscript that the Reviewer suggests.

“FASN has been previously related to relevant diverse features of tumor progression such as increased cell replication⁹, HER2-signaling¹⁰ or regulation of invadopodia¹¹.”

References

1. Schafer ZT, *et al.* Antioxidant and oncogene rescue of metabolic defects caused by loss of matrix attachment. *Nature* **461**, 109-113 (2009).
2. Anwar MR, Saldana-Caboverde A, Garcia S, Diaz F. The Organization of Mitochondrial Supercomplexes is Modulated by Oxidative Stress In Vivo in Mouse Models of Mitochondrial Encephalopathy. *Int J Mol Sci* **19**, (2018).
3. Lim SC, *et al.* Loss of the Mitochondrial Fatty Acid beta-Oxidation Protein Medium-Chain Acyl-Coenzyme A Dehydrogenase Disrupts Oxidative Phosphorylation Protein Complex Stability and Function. *Sci Rep* **8**, 153 (2018).
4. Goss PE, *et al.* Exemestane for breast-cancer prevention in postmenopausal women. *N Engl J Med* **364**, 2381-2391 (2011).

5. Cuzick J, *et al.* Anastrozole for prevention of breast cancer in high-risk postmenopausal women (IBIS-II): an international, double-blind, randomised placebo-controlled trial. *Lancet* **383**, 1041-1048 (2014).
6. Gregory MK, King HW, Bain PA, Gibson RA, Tocher DR, Schuller KA. Development of a fish cell culture model to investigate the impact of fish oil replacement on lipid peroxidation. *Lipids* **46**, 753-764 (2011).
7. Kuda O, *et al.* Sulfo-N-succinimidyl oleate (SSO) inhibits fatty acid uptake and signaling for intracellular calcium via binding CD36 lysine 164: SSO also inhibits oxidized low density lipoprotein uptake by macrophages. *J Biol Chem* **288**, 15547-15555 (2013).
8. Puig T, *et al.* Novel Inhibitors of Fatty Acid Synthase with Anticancer Activity. *Clin Cancer Res* **15**, 7608-7615 (2009).
9. Menendez JA, Lupu R. Fatty acid synthase and the lipogenic phenotype in cancer pathogenesis. *Nat Rev Cancer* **7**, 763-777 (2007).
10. Menendez JA, *et al.* Inhibition of fatty acid synthase (FAS) suppresses HER2/neu (erbB-2) oncogene overexpression in cancer cells. *Proc Natl Acad Sci U S A* **101**, 10715-10720 (2004).
11. Scott KE, *et al.* Metabolic regulation of invadopodia and invasion by acetyl-CoA carboxylase 1 and de novo lipogenesis. *PLoS One* **7**, e29761 (2012).

REVIEWERS' COMMENTS:

Reviewer #1 (Remarks to the Author):

Authors have mainly addressed the comments in their revised version of the manuscript. They significantly improved the quality and relevance of the study and discussed the crucial points raised in this publication.

One last point needs to be addressed, as previously asked but not properly corrected in the revised version to my point of view:

in figS3a authors have to indicate stats between FASNlox/lox-PyMT without Acetyl-CoA and FASNlox/lox-PyMT with Acetyl-CoA. Indeed they claim that inhibition of PDH activity observed in FASN Δ/Δ -PyMT compared with FASNlox/lox-PyMT is phenocopied by adding acetyl-CoA to FASNlox/lox-PyMT. We do observe a trend of decrease in FASNlox/lox-PyMT after addition of acetyl-CoA but illustrating stats would strengthen what's mentioned for this point in the result section.

Reviewer #2 (Remarks to the Author):

The authors properly addressed our concerns and performed the proposed experiments to support their claims. NAC/GSH rescue in spheroid formation on all genotypes studied (PyMT, Kras, HER2) was conducted and results were consistent. Sulfo-N-succinimidyl oleate (SSO) and albumin-complexed palmitate were not able to rescue colonies formation after FASN inhibition, strengthening the data suggesting that lack of palmitate is not the driver of the phenotype observed, a major finding for the role of FASN in cancer initiation and progression. FAO was properly assessed with determination of labeled ^{14}C – CO_2 . The results suggesting that part of intracellular lipid deposit is derived from exogenous lipid pool was strengthened by Oil Red O staining. Genetic silencing of IDH1 recapitulated FASN activity ablation, supporting the hypothesis that IDH1-driven reductive carboxylation plays a role in anchorage independence. Fatty-acid activity was properly evaluated using radiolabeled acetate and analyzing the labeled lipid pool. However some concepts remain overstated. The authors suggest FASN inhibition before transformation as preventive treatment, suggesting that long term “interventions should be planned in healthy patients at high risk of developing invasive breast cancer, such as those with previous diagnosis of atypical hyperplasia, lobular carcinoma in situ, or high 5-year risk score.” This should be toned down as it is based on data obtained mainly in MEFs. Also, the authors included in the discussion a reference that shows FASN regulation of invadopodia, but did not perform any experiment on invasive potential driven by reductive power of palmitate synthesis, rather than lack of fatty acids itself. There are several recent references in high profile journals on FASN inhibitors that are not cited

Rebuttal letter – NCOMMS-18-31933A

Reviewer #1:

Authors have mainly addressed the comments in their revised version of the manuscript. They significantly improved the quality and relevance of the study and discussed the crucial points raised in this publication.

One last point needs to be addressed, as previously asked but not properly corrected in the revised version to my point of view:

in figS3a authors have to indicate stats between FASNlox/lox-PyMT without Acetyl-CoA and FASNlox/lox-PyMT with Acetyl-CoA. Indeed they claim that inhibition of PDH activity observed in FASN Δ/Δ -PyMT compared with FASNlox/lox-PyMT is phenocopied by adding acetyl-CoA to FASNlox/lox-PyMT. We do observe a trend of decrease in FASNlox/lox-PyMT after addition of acetyl-CoA but illustrating stats would strengthen what's mentioned for this point in the result section.

We thank the reviewer for the time and dedication and her/his thoughtful comments; addressing those comments has improved significantly our work.

Regarding the last part of the comment, we apologize for the lack of clarity. Indeed, the experiment shows a trend towards a decrease in PDH activity in FASNlox/lox-PyMT. The resulting phenotype (FASNlox/lox-PyMT + Acetyl-CoA) displays a PDH activity that albeit higher than FASN Δ/Δ -PyMT, is not statistically significantly different – and thus that is why we said that it phenocopies the lack of FASN. Obviously there are more factors in PDH activity and at the end of the day this is an in vitro assay outside of the context of the cellular particular stoichiometry. In fact, FASNlox/lox-PyMT without and with Acetyl-CoA, again, are not statistically significantly different – although Acetyl-CoA decreases the activity of PDH, in vitro we only observe a trend (P value is

0.565, now added). Thus, we have toned-down the text in the main manuscript and state the observation as trends. The text in the result section is now as follows

“Using an *in vitro* assay with mitochondrial lysates (i.e., in conditions where the original mitochondrial stoichiometry is lost), we observed a trend to phenocopy such inhibition adding acetyl-CoA to FASN^{lox/lox}-PyMT lysates: pyruvate dehydrogenase activity was similar to that of FASN^{ΔΔ}-PyMT MEFS but not statistically significantly different – albeit lower – to that of FASN^{lox/lox}-PyMT MEFS (Supplementary Figure 3a)”

Reviewer #2:

The authors properly addressed our concerns and performed the proposed experiments to support their claims. NAC/GSH rescue in spheroid formation on all genotypes studied (PyMT, Kras, HER2) was conducted and results were consistent. Sulfo-N-succinimidyl oleate (SSO) and albumin-complexed palmitate were not able to rescue colonies formation after FASN inhibition, strengthening the data suggesting that lack of palmitate is not the driver of the phenotype observed, a major finding for the role of FASN in cancer initiation and progression. FAO was properly assessed with determination of labeled 14C – CO₂. The results suggesting that part of intracellular lipid deposit is derived from exogenous lipid pool was strengthened by Oil Red O staining. Genetic silencing of IDH1 recapitulated FASN activity ablation, supporting the hypothesis that IDH1-driven reductive carboxylation plays a role in anchorage independence. Fatty-acid activity was properly evaluated using radiolabeled acetate and analyzing the labeled lipid pool.

We thank the reviewer for the comments, her/his suggestions have significantly improved our work.

However some concepts remain overstated. The authors suggest FASN inhibition before transformation as preventive treatment, suggesting that long term “interventions should be planned in healthy patients at high risk of developing invasive breast cancer, such as those with previous diagnosis of atypical hyperplasia, lobular carcinoma in situ, or high 5-year risk score.” This should be toned down as it is based on data obtained mainly in MEFs.

We have modified the final paragraph following this recommendation. The text reads now as follows:

“Taken together, these features indicate FASN is a potential target for cancer prevention. Future studies with clinical-grade compounds in high-risk patients subpopulations (akin previous chemoprevention trials in breast cancer^{1,2}) could address the therapeutic utility of this strategy.”

Also, the authors included in the discussion a reference that shows FASN regulation of invadopodia, but did not perform any experiment on invasive potential driven by reductive power of palmitate synthesis, rather than lack of fatty acids itself.

We thank the reviewer for this comment. We had only included this reference due to previous request by the reviewer, but we agree that it adds nothing to the experimental results. It has been removed since we were already above the reference limit.

There are several recent references in high profile journals on FASN inhibitors that are not cited

Novel references have been included. The text is now as follows:

“With novel FASN inhibitors in perspective³⁻⁵, a clinical-grade compound that would selectively target”⁵

References

1. Goss, P.E. *et al.* Exemestane for breast-cancer prevention in postmenopausal women. *N Engl J Med* **364**, 2381-2391 (2011).
2. Cuzick, J. *et al.* Anastrozole for prevention of breast cancer in high-risk postmenopausal women (IBIS-II): an international, double-blind, randomised placebo-controlled trial. *Lancet* **383**, 1041-1048 (2014).
3. Kim, Y.C. *et al.* Toll-like receptor mediated inflammation requires FASN-dependent MYD88 palmitoylation. *Nature chemical biology* **15**, 907-916 (2019).
4. Jones, S.F. & Infante, J.R. Molecular Pathways: Fatty Acid Synthase. *Clin Cancer Res* **21**, 5434-5438 (2015).
5. Zadra, G. *et al.* Inhibition of de novo lipogenesis targets androgen receptor signaling in castration-resistant prostate cancer. *Proc Natl Acad Sci U S A* **116**, 631-640 (2019).